# Fed-Duet: Dual Expert-Orchestrated Framework for Continual Federated Vision-Language Learning

**Tao Guo**[1], **Junwei Chen**[1,2] **& Laizhong Cui**[1] *

[1]College of Computer Science and Software Engineering, Shenzhen University
[2]WeBank Instituite of Financial Technology, Shenzhen University
`cocogt@szu.edu.cn, 2023290205@email.szu.edu.cn, cuilz@szu.edu.cn`

## Abstract

Pretrained vision-language models (VLMs), such as CLIP, have shown promise in federated learning (FL) by bringing strong multimodal representations to edge devices. However, continual adaptation remains a core challenge in practical federated settings, where task distributions evolve over time and data remain non-IID across clients. In this emerging area, recent works adopt parameter-efficient fine-tuning (PEFT) as a lightweight way to reduce communication overhead, yet they fail to preserve satisfactory performance under continual learning conditions. Meanwhile, traditional federated continual learning (FCL) methods lack the capacity to maintain cross-modal alignment crucial to VLM performance. We introduce **Fed-Duet**, a novel **Du**al **E**xper**t**-orchestrated framework for efficient federated continual learning in vision-language models. Fed-Duet features a dual-expert adaptation mechanism, combining server-coordinated semantic prompts with client-personalized modular adapters. These pathways are dynamically fused via a cross-attention mechanism, enabling effective knowledge transfer while preserving multimodal alignment and mitigating forgetting. We evaluate Fed-Duet across multiple challenging continual learning tasks in federated vision-language settings and demonstrate that it achieves superior performance and stability compared to existing approaches. Our work highlights the importance of coordinated expert composition in enabling scalable and robust multimodal continual learning. The code is available at [FedDuet.](#)

## 1 Introduction

Federated Learning (FL) has emerged as a key paradigm for learning from sensitive and siloed data by enabling collaborative model training across decentralized data sources while preserving privacy (McMahan et al., 2017; Kairouz et al., 2021; Li et al., 2020; Zhao et al., 2018). Meanwhile, Large-scale Vision-Language Models (VLMs), pretrained on web-scale data, such as CLIP (Radford et al., 2021), have revolutionized multimodal AI by demonstrating unprecedented capabilities in zero-shot generalization and cross-modal understanding (Jia et al., 2021; Li et al., 2022; Du et al., 2022). Integrating VLMs into the FL paradigm offers great potential for empowering edge devices with advanced multimodal intelligence (Ren et al., 2025), while also introducing new challenges in downstream learning and adaptation.

The most immediate challenge in realizing this paradigm is the large-scale of modern VLMs. Their massive size makes the full-model fine-tuning required in each federated round prohibitively expensive due to immense communication costs. To surmount this bottleneck, Parameter-Efficient Fine-Tuning (PEFT) techniques (Houlsby et al., 2019; Lester et al., 2021; Feng et al., 2023; Li et al., 2024; Yu et al., 2024), such as adapter-based and prompt-based methods, have emerged as an essential solution. By freezing the vast VLM backbone and only training and communicating a small fraction of parameters, these methods drastically reduces the communication overhead (Lu et al., 2023; Guo et al., 2023b; 2024).

---

*Corresponding author.

Despite recent advances in efficiency, real-world edge environments remain highly challenging: *tasks evolve continuously and client data exhibit non-IID distributions.* (Kirkpatrick et al., 2017; Aljundi et al., 2019; Belouadah & Popescu, 2019; Wang et al., 2024). These dynamics give rise to the paradigm of Federated Continual Learning (FCL), which seeks to continually adapt models to new knowledge without catastrophic forgetting, while also addressing the inherent data heterogeneity of federated networks (Lopez-Paz & Ranzato, 2017; Zhao et al., 2020; Dong et al., 2022; So et al., 2022). A solution is therefore needed to address these challenges effectively.

However, directly applying traditional FCL methods to this new context is problematic because they are fundamentally ill-suited for VLMs. First, their reliance on a full-model update paradigm is computationally prohibitive for large-scale models and risks catastrophically overwriting pre-trained knowledge. Second, their uni-model perspective ignores the specialized, dual-stream nature of VLMs, thereby threatening the delicate cross-modal alignment that is central to their capabilities.

Furthermore, in the emerging field of federated vision-language model learning, most existing works have focused on applying PEFT techniques as a lightweight solution to reduce communication overhead, yet these methods introduce new challenges when applied under continual learning settings. 1) First, *applying a singular PEFT strategy leads to an adaptation imbalance*. Relying solely on high-level prompts may fail to capture client-specific nuances, while using only low-level adapters can weaken global semantic consistency. 2) Second, *aggregating sparse and heterogeneous PEFT updates across clients risks disrupting the VLMs' intrinsic cross-modal alignment*, which is essential for unified vision-language understanding. These challenges highlight the need for a coordinated orchestration mechanism that can effectively integrate diverse client adaptations while preserving the model's semantic integrity. This leads to our core research question:

*How can we design an orchestrated framework that efficiently addresses adaptation imbalance and cross-modal misalignment in federated vision-language learning?*

To overcome the above challenges, we propose **Fed-Duet**, a novel **Du**al-**E**xper**t** orchestrated framework for FCL in VLMs. Fed-Duet employs a dual-expert architecture that orchestrates two complementary pathways—*guiding prompts* for semantic guidance and *modular adapters* for fine-grained task specialization—whose outputs are dynamically integrated to ensure coherence and adaptability. Our main contributions are summarized as follows:

- We pioneer a paradigm shift in federated VLM learning by identifying a critical gap between the latest PEFT-based approaches and the need for continual adaptation as tasks evolve. By filling this gap, we unlock greater potential for retaining old knowledge while acquiring new insights, paving the way for more sustainable federated VLM learning.

- We propose Fed-Duet, a novel framework designed specifically for federated continual learning in CLIP-like VLMs under data heterogeneity. Fed-Duet enables effective knowledge transfer by synergizing two complementary expert pathways, while simultaneously maintaining multimodal alignment and mitigating forgetting.

- We validate the effectiveness of Fed-Duet through extensive experiments on challenging federated VLMs and FCL benchmarks. The superiority of our results offering a scalable solution for lifelong adaptation in federated vision-language settings.

## 2 RELATED WORK

### 2.1 CLIP-LIKE VLMS IN FEDERATED LEARNING

Pre-trained Vision-Language Models (VLMs) like CLIP (Radford et al., 2021) offer a promising foundation for Federated Learning. For instance, CLIP2FL (Shi et al., 2024) demonstrates this by leveraging a frozen CLIP model as a "teacher" to guide smaller models, specifically tackling data heterogeneity and long-tailed distributions. However, their massive size makes full-model aggregation infeasible due to prohibitive communication costs. To overcome this bottleneck, the community has adopted PEFT techniques, prominently featuring prompt-tuning and adapter-tuning (Houlsby et al., 2019; Lester et al., 2021; Li & Liang, 2021; Hu et al., 2022).

Prompt-tuning has emerged as the dominant PEFT paradigm for VLMs. The concept was popularized in centralized learning by methods like CoOp (Zhou et al., 2022b;a), which demonstrated

that optimizing a few learnable prompt vectors could effectively steer a frozen VLM. The communication efficiency of this approach made it a natural fit for FL. Building on this idea, subsequent works such as PromptFL (Guo et al., 2023b) and pFedPrompt (Guo et al., 2023a) have proposed federated prompt-tuning strategies, leveraging global aggregation and personalization, respectively. In parallel, adapter-tuning has shown promising results in federated settings, with methods like Fed-CLIP (Lu et al., 2023) demonstrating that communicating small, trainable adapter modules can also successfully adapt VLMs in a federated manner.

While these PEFT paradigms are effective in alleviating training or communicating costs, they have primarily focused on static FL scenarios. *This remains an open and challenging question of how to leverage and potentially synergize them in a continually evolving, non-stationary FCL environment.*

## 2.2 Federated Continual Learning

Early efforts in Federated Continual Learning (FCL) often involved directly adapting centralized Continual Learning (CL) strategies, such as rehearsal-based iCaRL (Rebuffi et al., 2017) or regularization-based LwF (Li & Hoiem, 2017) and EWC (Lee et al., 2017). However, as these methods were not designed to handle the statistical heterogeneity inherent to federated networks, they typically resulted in compromised global model performance.

This limitation motivated the development of native FCL frameworks with more sophisticated mechanisms. Foundational works like FedWeIT (Yoon et al., 2021) focused on parameter-level transfer, introducing a decomposition scheme where clients selectively re-weight sparse parameters from others to mitigate negative interference. Alternatively, FedKNOW (Luopan et al., 2023) proposed a client-centric, gradient-level solution that extracts critical model weights and integrates restored gradients to guide learning without interference. More recently, the paradigm has shifted towards prompt-based adaptation. Methods like Fed-CPrompt (Bagwe et al., 2023) demonstrate that communicating only lightweight prompts, guided by a dedicated contrastive loss to tackle heterogeneity and asynchronicity, can achieve excellent performance with high efficiency.

While these native FCL methods represent significant advances, their fundamentally uni-modal designs are ill-suited for VLMs. They risk disrupting the delicate cross-modal alignment that is essential to VLM capabilities. *This creates a critical need for a new FCL framework designed specifically to learn continually while preserving this vital cross-modal integrity.*

## 2.3 Mixture-of-Experts

The sparse Mixture-of-Experts (MoE) paradigm, matured through foundational works like the Switch Transformer (Fedus et al., 2022) and state-of-the-art models like GPT-4 (Achiam et al., 2023), has recently been applied to enhance centralized Continual Learning. For instance, the work by Yu et al. (Yu et al., 2024) successfully uses MoE-Adapters to learn sequential tasks in VLMs, employing a selector to preserve zero-shot capabilities. However, such centralized MoE-for-CL approaches are fundamentally ill-suited for the Federated Continual Learning setting. They would require communicating entire expert modules or complex selector mechanisms, leading to prohibitive communication costs and failing to address client data heterogeneity.

While some reseraches have attempted to incorporate MoE paradigm into FCL setting, to bridge the gap, such as MoAFCL (Zhang & Liu, 2025), they focus solely on applying MoE to server-side adapters, overlooking the crucial role of semantic guidance, limiting their adaptability in Non-IID scenarios. *This presents a critical research opportunity for a unified architecture that synergizes both semantic guidance and parametric specialization to create a robust, communication-efficient FCL framework, which our work aims to address.*

## 3 Methodology

**Preliminaries and Problem Formulation.** We consider a Federated Continual Learning setting composed of a central server and $C$ clients, $\mathcal{C} = \{c_1, \ldots, c_C\}$. Each client $c$ observes a private, sequential stream of tasks $\{D_c^t\}_{t=1}^{T_c}$. The objective in this challenging setting is for each client to learn an effective model, that performs well on its cumulative history of tasks by leveraging both shared knowledge from the federation and its own evolving, local experience.

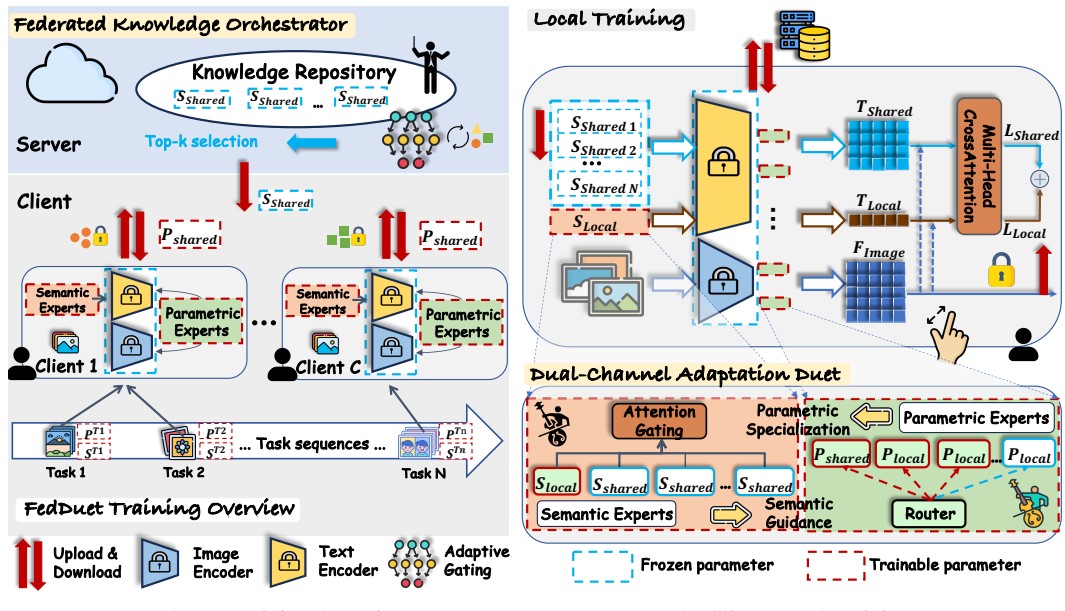

(a) FedDuet Training Overview         (b) Client Local Training

Figure 1: The architecture of **Fed-Duet**. **(a). FedDuet Overview, Interaction between Clients and Central Server.** The server-side *Federated Knowledge Orchestrator* maintains a Knowledge Repository and employs an adaptive gate to dispatch Shared Semantic Experts based on client features. **(b). Detailed Local Training.** The client-side *Dual-Expert Duet* adapts via two complementary pathways: the *Semantic Pathway* fuses Local and Shared experts via Cross-Attention Gating for semantic guidance, while the *Parametric Pathway* fine-tunes adapters for feature specialization.

**Framework Overview.** As illustrated in Figure 1, **Fed-Duet** orchestrates continual adaptation through two synergistic modules. 1) The server-side *Federated Knowledge Orchestrator* (Figure 1a) addresses knowledge coordination by leveraging a global repository and an adaptive gating mechanism to dispatch tailored Shared Semantic Experts to clients. 2) The client-side *Dual-Expert Duet* (Figure 1b) resolves adaptation imbalance via two parallel pathways: a *Semantic Pathway* dynamically fuses local and shared prompts through Cross-Attention Gating to provide semantic guidance, while a *Parametric Pathway* adapts modular experts for fine-grained feature specialization. This architecture is explicitly designed to preserve cross-modal alignment and is further regularized against catastrophic forgetting by two auxiliary losses, $\mathcal{L}_{\text{cross\_modal}}$ and $\mathcal{L}_{\text{stability}}$.

## 3.1 FEDERATED KNOWLEDGE ORCHESTRATOR

A key innovation of **Fed-Duet** lies in redefining the server's role from a simple aggregator to an intelligent knowledge orchestrator.

**The Orchestrated Knowledge Repository.** To ensure the Global Prompt Pool, $\mathcal{P} = \{\boldsymbol{p}_1, \ldots, \boldsymbol{p}_K\}$, is a semantically diverse repository from the outset, we forgo random initialization. Instead, we first derive $K$ conceptual anchors by performing K-Means clustering on the word embeddings of a large vocabulary (e.g., ImageNet-1k class names), yielding a set of centroids $\{\boldsymbol{c}_1, \ldots, \boldsymbol{c}_K\}$, where each $\boldsymbol{c}_k \in \mathbb{R}^D$.

Each prompt $\boldsymbol{p}_k \in \mathbb{R}^{L \times D}$ is then constructed using the textual template "a photo of [CLS]". The embeddings for the static context tokens ("a", "photo", "of") use their pre-trained values, while the embedding for the learnable [CLS] token is initialized directly with its corresponding centroid $\boldsymbol{c}_k$. This approach grounds each prompt in a meaningful semantic concept while providing a natural linguistic structure.

**Adaptive Dispatch Mechanism.** To enable efficient knowledge transfer without broadcasting the full repository, we introduce an Adaptive Gating network, $g_\theta$. This module learns to dispatch optimal experts based on a client's privacy-preserving feature summary, $\tilde{\boldsymbol{f}}_c$. The summary is computed as a global statistic derived through batch averaging, ensuring that no individual data is exposed.

Furthermore, as demonstrated in Sec. 4.2, this mechanism effectively preserves privacy while maintaining robustness under Differential Privacy (DP) noise injection. Guided by the client's training feedback, the gating is optimized via a loss-weighted binary cross-entropy (BCE) objective:

$$\mathcal{L}_{\text{gate}} = \sum_{c \in S_r} w_c \cdot \ell_{BCE}(g_\theta(\tilde{\boldsymbol{f}}_c), \boldsymbol{y}_c), \tag{1}$$

where $S_r$ is the participant set in round $r$. The weight $w_c = 1/(\mathcal{L}_c^{\text{final}} + \epsilon)$ prioritizes expert selections that yield lower client losses, ensuring the server learns to distribute the most effective knowledge.

## 3.2 DUAL-EXPERT DUET FOR CLIENT ADAPTATION

Our client-side module is built upon a **Du**al-**Expert** principle, consisting of two complementary adaptation pathways that disentangle semantic alignment from parametric feature transformation.

◇ **Semantic Experts Pathway.** This pathway enables semantic adaptation through learnable prompts, which function as semantic experts fused via a Dual-Stream Cross-Attention module. Given an input image feature, the model simultaneously attends to a private Local Semantic Expert that captures client-specific semantics and a set of server-provided Shared Semantic Experts representing generalizable semantics. The two sources yield separate expert logits (logits$_{\text{local}}$ and logits$_{\text{shared}}$), which are combined through a weighted fusion:

$$Logits_{final} = \lambda \cdot logits_{local} + (1 - \lambda) \cdot logits_{shared}. \tag{2}$$

This mechanism allows the model to dynamically balance personalized and shared semantic guidance on a per-sample basis (details in Appendix A.4).

◇ **Parametric Experts Pathway.** While prompt-based experts adjust semantic alignment, they do not directly transform internal features. To provide this complementary capability, we introduce an adapter-based pathway offering parametric experts for fine-grained representation adaptation. A server-aggregated Shared Adapter remains active to supply a stable and generalizable feature backbone, whereas private Local Adapters are selectively activated via a Top-$k$ router for efficient, on-demand personalization. This dual-path design forms dedicated parametric routes for robust generalization and targeted specialization, achieving balanced adaptation.

**Synergy of Dual Expert Pathways.** The two expert pathways reinforce each other in a bidirectional loop. The stable feature backbone established by the parametric pathway enables more precise semantic guidance from the semantic pathway, while the refined semantic signals provide clearer contextual cues that improve the routing of parametric experts. This continuous interplay between foundational stability and semantic refinement underpins the effectiveness of our framework.

## 3.3 OVERALL TRAINING STRATEGY

**Progressive Decoupled Optimization.** To resolve optimization conflicts and realize the bidirectional synergy between the two expert pathways, we introduce a Progressive Decoupled Optimization strategy. It first trains parametric experts exclusively to establish a stable feature foundation, then freezes them while training semantic experts to provide precise semantic guidance. This progressive scheme ensures that a stable parametric base empowers semantic guidance, which in turn improves subsequent parametric specialization.

## 3.4 SYNERGISTIC MULTI-OBJECTIVE LOSS

The client-side training is guided by a composite loss function, $\mathcal{L}_{\text{client}}$, designed to pursue multiple objectives simultaneously. The overall objective is defined as:

$$\mathcal{L}_{\text{client}} = \mathcal{L}_{CE} + \alpha \mathcal{L}_{\text{moe}} + \eta \mathcal{L}_{\text{cross\_modal}} + \gamma \mathcal{L}_{\text{stability}}. \tag{3}$$

where the primary objective is standard cross-entropy loss ($\mathcal{L}_{CE}$), supplemented by a load-balancing loss ($\mathcal{L}_{\text{moe}}$) for the experts, as well as two auxiliary losses, $\mathcal{L}_{\text{cross\_modal}}$ and $\mathcal{L}_{\text{stability}}$ to address key FCL challenges. Specifically, we detail the two loss components as follows:

**Routing Consistency Loss ($\mathcal{L}_{\text{cross\_modal}}$) for Alignment.** To counteract the tendency of standard MoE layers to disrupt a VLM's inherent vision-language alignment, we introduce a routing consistency loss inspired by CLIP's contrastive objective. This loss enforces multimodal alignment by

ensuring the expert routing for an image is consistent with that of its corresponding text. The loss is formulated as:

$$\mathcal{L}_{\text{cross\_modal}} = \frac{1}{2} \left( \text{CE}(\mathbf{S}/\tau, \mathbf{y}) + \text{CE}(\mathbf{S}^{\top}/\tau, \mathbf{y}) \right), \tag{4}$$

where $\mathbf{S}$ is a similarity matrix computed from the routing distributions of image-text pairs within a batch, $\tau$ is a temperature hyperparamete that controls the sharpness of the distribution, and $\mathbf{y}$ represents the ground-truth labels indicating the correct image-text pairings. The symmetric cross-entropy objective guides the experts toward learning modality-invariant representations.

**Expert Stability Loss ($\mathcal{L}_{\textbf{stability}}$) for Anti-Forgetting.** To mitigate catastrophic forgetting, we employ an expert stability loss that acts as a form of knowledge distillation on the routing policy. This loss preserves learned routing behaviors by regularizing the current policy to remain close to historical ones. It is defined using KL divergence as:

$$\mathcal{L}_{\text{stability}} = D_{KL}(\mathbf{p}^{(t)} || \bar{\mathbf{p}}^{(t-1)}), \tag{5}$$

where $\mathbf{p}^{(t)}$ is the expert routing distribution for the current task and $\bar{\mathbf{p}}^{(t-1)}$ is the historical policy from previous tasks, maintained on a per-layer basis using an exponential moving average. This regularization provides a robust mechanism against forgetting at the expert level.

## 4 EXPERIMENTS

### 4.1 EXPERIMENTAL SETTINGS

**Datasets and Task Settings.** We evaluate our framework on three benchmarks: CIFAR-100 (Krizhevsky & Hinton, 2009) and Tiny-ImageNet (Le & Yang, 2015) for Class-Incremental Learning (partitioned into 5 and 10 tasks), and DomainNet (Peng et al., 2019) for Domain-Incremental Learning. The DomainNet setup, where each domain constitutes a new task, rigorously tests the model's robustness against distribution shifts.

**Evaluation Metrics.** Overall performance is measured by *Average Accuracy* and *Last Accuracy*, while knowledge retention is captured by the *Forgetting Measure*. To assess the stability–plasticity trade-off, we report *Stability*, *Plasticity*, and *Continual Utility*. Cross-modal alignment is evaluated via *Recall@K* and *Alignment Score*. Finally, to rigorously assess defense against privacy attacks, we employ *Peak Signal-to-Noise Ratio (PSNR)* and *Structural Similarity Index (SSIM)* to quantify image reconstruction quality. Formal definitions are provided in Appendix A.3.1.

**Baselines.** To comprehensively evaluate our framework, we compare it against a range of benchmarks from three main categories: 1) **Standard FL Methods**, foundational approaches not designed for continual learning, used to establish a performance baseline (e.g., FedAvg (McMahan et al., 2017) and FedProx (Li et al., 2020)); 2) **Representative FCL Benchmarks**, which include regularization-based approaches (Fed-EWC (Lee et al., 2017), Fed-LwF (Li & Hoiem, 2017)), a parameter-transfer method (FedWeIT (Yoon et al., 2021)), and a gradient-integration approach (Fed-KNOW (Luopan et al., 2023)); 3) **SOTA PEFT-based FCL Approaches for VLMs**, methods that also leverage Parameter-Efficient Fine-Tuning method for FCL with large-scale models. This category includes prompt-based methods like Fed-CPrompt (Bagwe et al., 2023), pFedMoAP (Luo et al., 2024), a Mixture-of-Experts framework where clients download multiple prompts as experts, and Powder (Piao et al., 2024), which facilitates dual knowledge transfer across clients and tasks using a task correlation matrix to guide prompt aggregation. This group is rounded out by the Mixture-of-Adapter based MoAFCL (Zhang & Liu, 2025). In addition to the FCL methods, our benchmark includes FedCLIP (Lu et al., 2023), a strong PEFT-based method for static Federated VLMs setting.

Furthermore, to ensure a fair comparison against our VLM-native approach, we enhance key baselines by uniformly equipping them with a CLIP backbone, following the protocol of MoAFCL. This includes methods like FedWeIT, FedKNOW, and other ViT-based approaches such as Fed-CPrompt and Powder. This creates a level playing field for all comparisons.

**Implementation Details.** Following previous research, we employ a pre-trained CLIP model as the VLM backbone across all experiments. Our federated system consists of one central server and five clients. Each experiment is repeated three times with different random seeds (42, 2005, 2026) and we report the averaged outcomes. We use the Adam optimizer with a learning rate of 3e-5. All experiments were conducted on a single NVIDIA RTX 4090 GPU with 24GB of VRAM.

Table 1: The performance of our method against SOTA benchmarks on representative CIFAR-100 and Tiny-ImageNet datasets under various FCL settings. Non-IID ($\beta$) indicates the Dirichlet parameter is set to $\beta$, Avg denotes the average accuracy over all learned tasks, while Last denotes the accuracy on the final task. Best results are in bold. The results show the superiority of our method.

| Data partition | | IID | | | | Non-IID ($\beta = 0.5$) | | | | Non-IID ($\beta = 0.1$) | | | |
|---|---|---|---|---|---|---|---|---|---|---|---|---|---|
| Tasks | | T=5 | | T=10 | | T=5 | | T=10 | | T=5 | | T=10 | |
| Dataset | Method | Avg ↑ | Last ↑ | Avg ↑ | Last ↑ | Avg ↑ | Last ↑ | Avg ↑ | Last ↑ | Avg ↑ | Last ↑ | Avg ↑ | Last ↑ |
| CIFAR-100 | FedAvg | 46.56 | 21.65 | 32.86 | 10.67 | 35.57 | 15.67 | 20.54 | 5.86 | 16.90 | 5.71 | 10.58 | 2.06 |
| | FedProx | 48.33 | 23.54 | 37.17 | 14.18 | 38.84 | 17.68 | 29.17 | 12.74 | 19.46 | 10.23 | 12.49 | 2.58 |
| | Fed-EWC | 49.17 | 23.54 | 38.18 | 13.18 | 38.56 | 18.48 | 26.78 | 10.22 | 28.34 | 22.48 | 11.98 | 3.18 |
| | Fed-LwF | 53.69 | 27.21 | 41.17 | 17.88 | 39.41 | 19.77 | 30.92 | 13.24 | 21.25 | 12.19 | 16.18 | 6.94 |
| | FedWeIT | 72.52 | 57.52 | 72.53 | 62.67 | 71.54 | 56.98 | 72.06 | 61.55 | 70.43 | 54.59 | 71.58 | 61.58 |
| | FedKNOW | 78.47 | 74.19 | 79.27 | 73.59 | 77.34 | 73.07 | 78.76 | 72.56 | 77.25 | 72.95 | 77.55 | 72.16 |
| | FedCLIP | 78.21 | 70.12 | 78.33 | 68.86 | 76.41 | 68.39 | 77.61 | 68.98 | 74.17 | 65.41 | 75.97 | 66.41 |
| | Fed-CPrompt | 73.41 | 65.87 | 74.58 | 65.71 | 73.07 | 65.69 | 74.39 | 65.03 | 73.00 | 65.27 | 74.20 | 65.16 |
| | Powder | 73.32 | 65.81 | 74.45 | 66.04 | 73.11 | 65.79 | 74.42 | 65.42 | 72.71 | 65.75 | 73.79 | 65.81 |
| | pFedMoAP | 76.61 | 68.65 | 76.80 | 67.86 | 70.02 | 62.79 | 70.19 | 60.16 | 52.63 | 47.57 | 58.46 | 50.61 |
| | MoAFCL | 77.93 | 69.05 | 77.72 | 65.84 | 73.21 | 64.72 | 75.51 | 65.27 | 65.96 | 59.78 | 68.47 | 60.73 |
| | **Fed-Duet (Ours)** | **86.21** | **79.11** | **86.22** | **78.56** | **84.77** | **78.48** | **84.55** | **77.97** | **84.58** | **75.97** | **84.22** | **75.88** |
| Tiny-ImageNet | FedAvg | 40.34 | 17.80 | 27.84 | 9.08 | 29.83 | 12.40 | 19.76 | 5.83 | 14.52 | 5.54 | 9.44 | 2.67 |
| | FedProx | 43.41 | 20.34 | 31.05 | 11.47 | 33.75 | 15.36 | 22.79 | 6.87 | 17.57 | 8.64 | 12.84 | 3.51 |
| | Fed-EWC | 43.07 | 21.16 | 29.85 | 9.87 | 31.57 | 13.59 | 27.37 | 9.57 | 22.73 | 10.71 | 16.57 | 4.64 |
| | Fed-LwF | 42.76 | 17.58 | 31.05 | 27.18 | 32.27 | 14.21 | 24.08 | 7.86 | 19.59 | 9.39 | 15.27 | 5.47 |
| | FedWeIT | 72.06 | 62.98 | 72.03 | 62.84 | 71.38 | 62.31 | 71.51 | 62.78 | 71.06 | 62.25 | 70.45 | 61.73 |
| | FedKNOW | 76.24 | 71.79 | 77.68 | 71.58 | 75.32 | 70.77 | 76.82 | 70.86 | 74.61 | 70.01 | 75.68 | 70.18 |
| | FedCLIP | 76.14 | 68.59 | 76.67 | 67.91 | 74.53 | 66.79 | 75.59 | 66.14 | 71.94 | 63.76 | 73.57 | 63.61 |
| | Fed-CPrompt | 73.09 | 65.75 | 74.27 | 65.79 | 72.55 | 65.93 | 74.20 | 65.41 | 72.48 | 65.47 | 74.16 | 65.31 |
| | Powder | 72.98 | 65.82 | 74.26 | 65.95 | 72.77 | 65.54 | 74.11 | 65.71 | 72.71 | 65.75 | 73.95 | 65.18 |
| | pFedMoAP | 74.47 | 67.28 | 75.06 | 66.29 | 69.96 | 61.72 | 70.42 | 61.46 | 55.47 | 48.45 | 57.87 | 52.42 |
| | MoAFCL | 74.16 | 65.30 | 74.17 | 64.24 | 70.27 | 62.17 | 71.39 | 61.52 | 62.51 | 59.47 | 66.84 | 59.33 |
| | **Fed-Duet (Ours)** | **82.60** | **77.44** | **83.52** | **75.85** | **81.32** | **75.21** | **81.75** | **74.81** | **80.48** | **74.03** | **81.56** | **73.57** |

## 4.2 OVERALL PERFORMANCE

**Performance Comparison.** As shown in Table 1, Fed-Duet demonstrates clear superiority across three key dimensions. 1) *Absolute Accuracy:* It achieves the highest final accuracy, outperforming the strongest baseline (FedKNOW Luopan et al. (2023)) by a significant 6.67% in the challenging CIFAR-100 scenario ($\beta = 0.1$, T=10). 2) *Data Heterogeneity Stability:* Fed-Duet is exceptionally robust to Non-IID data. While the performance of advanced methods like pFedMoAP (Luo et al., 2024) degrades by 24% under severe heterogeneity, our method's accuracy drops by a mere 2%. 3) *Continual Learning Stability:* As visualized in Figure 2, Fed-Duet also shows superior resistance to catastrophic forgetting, maintaining a high and stable performance trajectory as new tasks are introduced. These advantages validate that our dual-expert architecture effectively balances client specialization with robust knowledge retention.

**Evaluation on Domain Shift.** To assess robustness against severe domain shifts, we evaluate our method on DomainNet in a domain-incremental setting. As shown in Table 2, Fed-Duet achieves new SOTA performance, with an average accuracy of 68.47%. This marks a substantial improvement of 5.64% over the strongest baseline. This result underscores the effectiveness of our dual-expert adaptation, which preserves general knowledge while specializing in new visual domains to achieve strong generalization.

Table 2: Performance comparison on DomainNet. Best results are in bold. The blue values ($\Delta$) indicate a score lower than our method. Fed-Duet demonstrates strong domain generalization capabilities.

| Method | Avg Acc ↑ | $\Delta$ | Last Acc ↑ | $\Delta$ |
|---|---|---|---|---|
| FedWeIT | 62.20 | -6.27 | 58.54 | -7.51 |
| FedKNOW | 56.09 | -12.38 | 54.25 | -11.80 |
| FedCLIP | 62.83 | -5.64 | 60.04 | -6.01 |
| Fed-CPrompt | 56.16 | -12.31 | 54.38 | -11.67 |
| Powder | 56.14 | -12.33 | 54.35 | -11.70 |
| pFedMoAP | 59.98 | -8.49 | 56.35 | -9.70 |
| MoAFCL | 60.92 | -7.55 | 52.52 | -13.53 |
| **Fed-Duet (Ours)** | **68.47** | – | **66.05** | – |

**Evaluation on Cross-Modal Alignment.** Table 3 shows that stronger cross-modal alignment leads to clear gains in retrieval accuracy. After the final task, FedDuet achieves the best I2T and T2I Re-

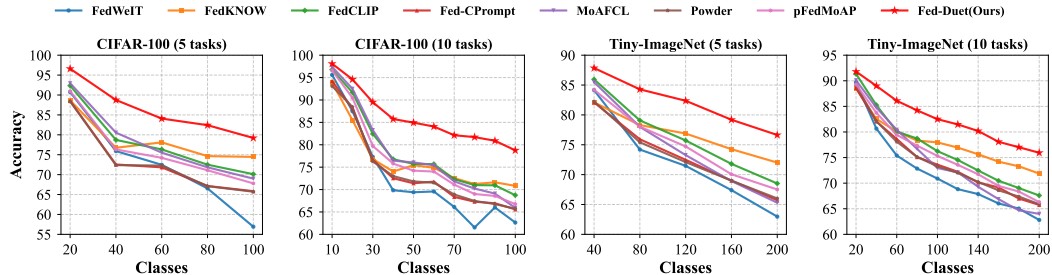

Figure 2: Performance trajectory of our method compared to SOTA benchmarks as the number of classes increases. The plots show accuracy as a function of the number of learned classes under 5 or 10 task scenarios. Our method, Fed-Duet, consistently outperforms all baselines, demonstrating superior resistance to catastrophic forgetting as tasks accumulate. Additional experimental results under various settings are presented in Appendix A.5 with details.

Table 3: Cross-modal retrieval performance on CIFAR-100. Metrics include image-to-text (I2T) and text-to-image (T2I) Recall@K. Best results are in **bold**. The blue values ($\Delta$) indicate a score lower than our method. Results are averaged across 5 clients.

| Method | I2T | | | T2I | | | Avg. | $\Delta$ |
|---|---|---|---|---|---|---|---|---|
| | **R@1** | **R@5** | **R@10** | **R@1** | **R@5** | **R@10** | | |
| FedWeIT | 64.02 | 86.47 | 92.17 | 84.00 | 99.60 | 100.0 | 87.71 | -4.74 |
| FedKNOW | 64.02 | 86.47 | 92.17 | 84.00 | 99.60 | 100.0 | 87.71 | -4.74 |
| FedCLIP | 63.96 | 86.69 | 91.76 | 86.60 | 99.00 | 99.80 | 87.97 | -4.48 |
| Fed-CPrompt | 63.81 | 86.74 | 91.91 | 84.40 | 98.60 | 100.0 | 87.58 | -4.87 |
| Powder | 63.91 | 86.94 | 91.82 | 85.80 | 99.40 | 100.0 | 87.98 | -4.47 |
| pFedMoAP | 63.66 | 87.02 | 91.95 | 87.00 | 99.00 | 100.0 | 88.11 | -4.34 |
| MoAFCL | 63.68 | 86.69 | 91.89 | 85.80 | 99.20 | 99.60 | 87.81 | -4.64 |
| **FedDuet (Ours)** | **78.18** | **96.13** | **98.33** | **93.20** | **99.80** | **100.0** | **92.45** | – |

call@K, outperforming the strongest baseline by +13.16% (I2T R@1) and +6.20% (T2I R@1). The consistent improvements across all Recall@K metrics indicate that FedDuet learns more discriminative and semantically coherent representations, resulting in better cross-modal generalization under federated continual learning.

Table 4 further explains these improvements. Baseline methods maintain nearly unchanged alignment scores around 0.06 across tasks, revealing limited ability to preserve cross-modal consistency. In contrast, FedDuet achieves an average alignment of 0.2003 on CIFAR-100, which corresponds to more than a 3× improvement compared with the typical baseline level. This substantial and stable margin indicates that our decoupled optimization strategy effectively mitigates forgetting while preserving semantic coherence between modalities throughout the learning process.

Table 4: Cross-modal alignment scores across tasks on CIFAR-100. Best results are in **bold**. The blue values ($\Delta$) indicate a score lower than our method. Results are averaged across 5 clients.

| Method | Task 0 | Task 1 | Task 2 | Task 3 | Task 4 | Avg. | $\Delta$ |
|---|---|---|---|---|---|---|---|
| FedWeIT | 0.0623 | 0.0578 | 0.0613 | 0.0610 | 0.0608 | 0.0606 | -0.1397 |
| FedKNOW | 0.0623 | 0.0578 | 0.0613 | 0.0610 | 0.0608 | 0.0606 | -0.1397 |
| FedCLIP | 0.0625 | 0.0580 | 0.0615 | 0.0611 | 0.0609 | 0.0608 | -0.1395 |
| Fed-CPrompt | 0.0625 | 0.0579 | 0.0615 | 0.0611 | 0.0608 | 0.0608 | -0.1395 |
| Powder | 0.0625 | 0.0576 | 0.0613 | 0.0610 | 0.0610 | 0.0607 | -0.1396 |
| pFedMoAP | 0.0626 | 0.0579 | 0.0614 | 0.0611 | 0.0609 | 0.0608 | -0.1395 |
| MoAFCL | 0.0621 | 0.0580 | 0.0615 | 0.0610 | 0.0609 | 0.0607 | -0.1396 |
| **FedDuet (Ours)** | **0.1929** | **0.1825** | **0.2140** | **0.2061** | **0.2061** | **0.2003** | – |

**Analysis of Privacy Compatibility and Robustness.** To validate Fed-Duet under rigorous Differential Privacy (DP) standards, we evaluate the trade-off between utility and defense by simulating noise injection (details in Appendix A.4.2). 1) Utility Preservation (Table 5a): Fed-Duet exhibits exceptional resilience, maintaining negligible accuracy degradation ($< 0.3\%$) even under high-noise regimes ($\sigma = 10$). 2) Defense against Reconstruction (Table 5b): Gradient-based attacks confirm that our aggregation mechanism inherently precludes reconstruction. Crucially, the reconstruction metrics for raw features ($\sigma = 0$) are statistically comparable to those under strict DP noise ($\sigma = 10$), with both yielding near-zero SSIM ($\approx 0.01$) and low PSNR ($< 9$ dB). This demonstrates that our inherent feature provides a primary and sufficient privacy guarantee.

Table 5: Quantitative results for privacy compatibility experiments. **(a)** Classification accuracy under varying Laplacian noise scales ($\sigma$). **(b)** Image quality metrics (PSNR and SSIM) derived from gradient-based reconstruction attacks at different optimization steps.

(a) Accuracy under Laplacian Noise ($\sigma$)

| $\sigma$ | CIFAR-100 | | Tiny-ImageNet | |
|---|---|---|---|---|
| | Avg | Last | Avg | Last |
| 0 | 86.21 | 79.11 | 82.60 | 77.44 |
| 0.1 | 85.96 | 78.86 | 82.55 | 76.17 |
| 1.0 | 85.94 | 78.51 | 82.53 | 76.11 |
| 10.0 | 85.91 | 78.48 | 82.52 | 76.25 |

(b) Reconstruction Attack Metrics

| Iter. | No DP ($\sigma = 0$) | | Noisy ($\sigma = 10$) | |
|---|---|---|---|---|
| | PSNR | SSIM | PSNR | SSIM |
| 200 | 7.62 | 0.034 | 7.05 | 0.017 |
| 600 | 8.14 | 0.012 | 7.08 | 0.008 |
| 800 | 8.11 | 0.013 | 8.04 | 0.008 |
| **1000** | **8.43** | **0.015** | **7.52** | **0.008** |

## 4.3 ABLATION STUDY

**Analysis on Core Components.** We conduct detailed ablation studies of different components on our framework, as shown in Table 6, from two central innovations of our framework: the *dual-expert architecture* and the *composite loss function*. 1) First, we evaluate the architecture by decoupling its pathways, which confirms our adaptation imbalance premise. The semantic-only pathway (Base-w/o PE) is unstable and suffers from high forgetting, while the parametric-only pathway (Base-w/o SE), though more stable, remains suboptimal. In contrast, our dual-expert Fed-Duet (Base) model substantially boosts accuracy, proving the synergistic benefit of combining semantic and parametric pathways. 2) Next, We analyze our auxiliary losses against advanced prompt-based FCL methods. While our Fed-Duet (Base) model is highly accurate, it exhibits a higher forgetting score than prompt-based FCL methods. This trade-off is systematically resolved by our two complementary auxiliary losses: the expert stability loss primarily mitigates forgetting, while the routing consistency loss enhances accuracy by enforcing multimodal alignment in the expert representations. The final Fed-Duet (Full) model, benefiting from their synergy, achieves both the highest accuracy and the lowest forgetting score, outperforming the baseline and validating the contribution of our composite loss design.

Table 6: We analyze the performance contribution of Fed-Duet's core components separately. Best results are in bold. The $\Delta$ values denote the performance difference from the full model (blue for lower, red for higher).

| Method | Avg Acc ↑ | $\Delta$ | Forget ↓ | $\Delta$ |
|---|---|---|---|---|
| Fed-CPrompt | 72.98 | -7.45 | 7.95 | +0.13 |
| Powder | 72.60 | -7.83 | 8.51 | +0.69 |
| pFedMoAP | 55.12 | -25.31 | 8.28 | +0.46 |
| Fed-Duet (Base-w/o PE) | 64.34 | -16.09 | 11.89 | +4.07 |
| Fed-Duet (Base-w/o SE) | 70.64 | -9.79 | 8.89 | +1.07 |
| Fed-Duet (Base) | 77.96 | -2.47 | 9.22 | +1.40 |
| Fed-Duet (Base + $\mathcal{L}_{cross\_modal}$) | 79.09 | -1.34 | 8.96 | +1.14 |
| Fed-Duet (Base + $\mathcal{L}_{stability}$) | 79.46 | -0.97 | 8.02 | +0.20 |
| **Fed-Duet (Full)** | **80.43** | – | **7.82** | – |

**Analysis of the Stability-Plasticity.** Our framework effectively resolves the core stability-plasticity dilemma. As shown in Figure 3a, Fed-Duet simultaneously achieves both the highest stability and plasticity scores, a superior balance that contrasts sharply with leading baselines that are forced to compromise one for the other. To formally quantify this trade-off, we introduce the Continual Utility score, a unified metric representing the weighted aggregation of Stability and Plasticity. This robust balance is further validated by our framework's continual utility score, as shown in Figure 3b. Our method consistently dominates the top score, maintaining a significant performance lead across the

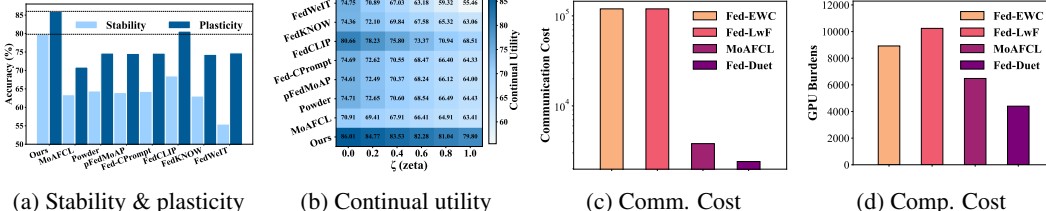

(a) Stability & plasticity  (b) Continual utility  (c) Comm. Cost  (d) Comp. Cost

Figure 3: Analysis of the stability-plasticity trade-off and efficiency on CIFAR-100 with 5 incremental tasks. **(a)** Stability and plasticity scores compared with different methods. Our method excels in both aspects. **(b)** The continual utility trade-off for our method across varying values of the hyperparameter $\zeta$. A larger $\zeta$ indicates more emphasis on stability. Darker colors indicate better utility. **(c)** Total communication cost (MB, log scale) required by each method. **(d)** Peak GPU memory footprint (MiB) during local client training. Lower values correspond to higher efficiency.

entire range of the trade-off hyperparameter $\zeta$. This result shows that Fed-Duet not only establishes a superior balance but also demonstrates remarkable robustness to hyperparameter choices.

**Efficiency Analysis.** Our framework offers significant advantages in both communication and computational efficiency. As shown in Figure 3c, Fed-Duet is not only orders of magnitude more efficient than full-parameter tuning methods like Fed-EWC and FedLwF but also reduces overhead by approximately another order of magnitude relative to the PEFT-based MoAFCL. A rigorous baseline selection is crucial, as existing methods are largely incompatible with our research context. For instance, pFedMoAP (Luo et al., 2024) is not designed for continual learning, while Powder (Piao et al., 2024) utilizes a standard ViT backbone. We therefore selected MoAFCL (Zhang & Liu, 2025) as our primary baseline since it is the only framework specifically engineered for the federated continual learning of VLMs like CLIP. This efficiency extends to computation, measured by GPU memory burdens in Figure 3d, where Fed-Duet again achieves the lowest footprint. This demonstrates that our dual-expert architecture provides a highly resource-efficient solution for FCL, enhancing continual learning while significantly reducing both communication and computation burdens.

**Analysis of Large-Scale Scalability.** To rigorously assess our framework's scalability, we extended the experimental setup to a more demanding 20-client environment. The performance trajectories, visualized in Figure 4, reveal two critical insights into Fed-Duet's robustness. Firstly, Fed-Duet occupies the highest accuracy plane across all 10 tasks, consistently surpassing all baseline methods. Secondly, its performance slope is notably flatter, demonstrating superior resilience against the performance degradation that hinders the baselines. This visual evidence substantiates that Fed-Duet's core mechanism for mitigating catastrophic forgetting remains potent at scale, ensuring sustained high performance in larger and more complex federated systems.

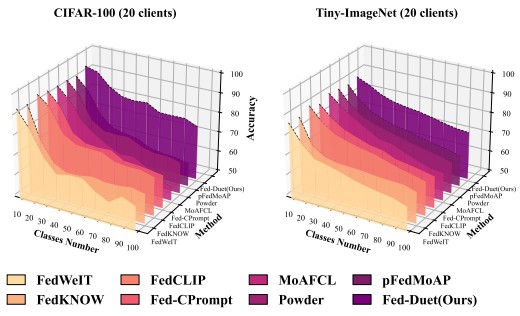

Figure 4: Performance trajectory with large-scale (20-client) setting. Accuracy (z-axis) is plotted across 10 sequential tasks on CIFAR-100 and Tiny-ImageNet. Fed-Duet remains robust under large-scale scenario.

## 5  CONCLUSION

We present Fed-Duet, a framework addressing Federated Continual Learning for Vision-Language Models. By synergizing a server-side *Knowledge Orchestrator* with client-side *Dual-Expert Pathways*, our approach decouples semantic guidance from parametric specialization, effectively resolving the plasticity-stability dilemma. Extensive experiments confirm that Fed-Duet achieves SOTA performance in accuracy and cross-modal alignment. Furthermore, its proven compatibility with rigorous differential privacy standards validates it as a scalable, secure solution for deploying foundation models in dynamic edge environments.

ETHICS STATEMENT.

This paper adheres to the ICLR Code of Ethics. The research presented in this paper does not involve human participants, animals, or any other ethical considerations. All data used in this study were obtained from publicly available sources. Our method does not involve discrimination, bias, or fairness concerns.

REPRODUCIBILITY STATEMENT.

To ensure reproducibility, we provide our source code, running scripts, and hyperparameter configurations. We also include detailed algorithm pseudocode, dataset information, and additional experimental results in Appendix A.5 to facilitate the verification of our findings.

ACKNOWLEDGMENTS.

This work has been partially supported by National Natural Science Foundation of China under Grant No. U23B2026 and No.62372305, Guangdong Basic and Applied Basic Research Foundation under Grant No. 2024B1515040012, Shenzhen Science and Technology Program under Grant No. KJZD20230923114809020, and Research Team Cultivation Program of Shenzhen University, Grant No.2023QNT015.

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

# A APPENDIX

## A.1 ALGORITHM PSEUDOCODE

We provide the detailed pseudocode for the Fed-Duet framework. The overall process is divided into the server-side orchestration (Algorithm 1) and the client-side dual-expert training (Algorithm 2). This describes the federated learning process for a single continual learning task.

---

**Algorithm 1** Fed-Duet: Server-Side Procedure

---

1: **Initialize:** Global Prompt Pool $\mathcal{P}$, Adaptive Gating Net $g_\phi$.
2: **Initialize:** Global Shared Expert $\boldsymbol{E}_{\text{shared}}$.
3: **for** each communication round $r = 1, 2, ..., R$ **do**
4:     Select a subset of clients $\mathcal{S}_r$.
5:     Initialize GateNet buffer $\mathcal{B} \leftarrow \varnothing$.
6:     **for** each client $c \in \mathcal{S}_r$ **in parallel do**
7:         Receive feature summary $\boldsymbol{f}_c$ from client $c$.
8:         **if** $\boldsymbol{f}_c$ is available **then**
9:             Select prompt indices $\mathbf{idx}_c \leftarrow \text{TopK}_i(g_\phi(\boldsymbol{f}_c))$.
10:         **else**
11:             {Cold Start}
12:             Select random prompt indices $\mathbf{idx}_c$.
13:         **end if**
14:         Send selected prompts $\{\mathcal{P}[i]\}_{i \in \mathbf{idx}_c}$ and $\boldsymbol{E}_{\text{shared}}$ to client $c$.
15:         Receive updated shared expert $\boldsymbol{E}_c^{\text{shared}}$ and final loss $\mathcal{L}_c$.
16:         Add tuple $(\boldsymbol{f}_c, \mathbf{idx}_c, \mathcal{L}_c)$ to buffer $\mathcal{B}$.
17:     **end for**
18:     **Aggregate Shared Experts:**
19:     $\boldsymbol{E}_{\text{shared}} \leftarrow \sum_{c \in \mathcal{S}_r} \omega_c \boldsymbol{E}_c^{\text{shared}}$ {$\omega_c$ is client weight}
20:     **Train Gating Net:**
21:     Compute loss $\mathcal{L}_{\text{gate}} = \sum_{(\boldsymbol{f}_c, \mathbf{idx}_c, \mathcal{L}_c) \in \mathcal{B}} \frac{1}{\mathcal{L}_c + \epsilon} \cdot \text{BCE}(g_\phi(\boldsymbol{f}_c), \mathbf{1}_{\mathbf{idx}_c})$.
22:     Update $g_\phi$ by descending the gradient of $\mathcal{L}_{\text{gate}}$.
23: **end for**

---

## A.2 DATASET AND TASK SETTINGS

We evaluate our framework under three distinct continual learning paradigms, each designed to test different facets of model performance. The statistics for all datasets are summarized in Table 7.

Table 7: Statistics of the datasets used in our experiments.

| Dataset | Classes | Train | Test |
|---|---|---|---|
| CIFAR-100 | 100 | 50,000 | 10,000 |
| Tiny-ImageNet | 200 | 100,000 | 10,000 |
| DomainNet | 345 | 586,576 (Total) | |
| Flowers102 | 102 | 2,040 | 6,149 |
| Oxford Pets | 37 | 3,680 | 3,669 |
| Food101 | 101 | 75,750 | 25,250 |
| Caltech-101 | 101 | 3,000 | 6,000 |
| DTD | 47 | 3,760 | 1,880 |

**Class-Incremental Learning on CIFAR-100 and Tiny-ImageNet.** For this setting, the model must learn new classes over time without forgetting old ones.

- **Task Construction:** For CIFAR-100 (100 classes), we create scenarios with 5 sequential tasks of 20 classes each, and 10 tasks of 10 classes each. For Tiny-ImageNet (200 classes),

---

**Algorithm 2** Fed-Duet: Client-Side Procedure

---

1: **Input:** Shared prompts $\mathcal{P}_s$, shared expert $\boldsymbol{E}_{\text{shared}}$, current round $r$.
2: **Initialize:** Local prompt $\boldsymbol{p}^{\text{local}}$, local experts $\boldsymbol{E}^{\text{local}}$.
3: Compute privacy-preserving feature summary $\boldsymbol{f}_c$ from local data $D_c^t$.
4: **if** $r \leq R/2$ **then**
5:     {**Stage 1: Adapt Parametric Experts**}
6:     **for** local epoch $e = 1, ..., E$ **do**
7:         **for** batch $(x, y)$ from $D_c^t$ **do**
8:             Compute features using MoE-Adapter (combining $\boldsymbol{E}_{\text{shared}}$ and $\boldsymbol{E}^{\text{local}}$).
9:             Compute loss $\mathcal{L} = \mathcal{L}_{\text{CE}} + \alpha \mathcal{L}_{\text{moe}} + \beta \mathcal{L}_{\text{cross}} + \gamma \mathcal{L}_{\text{stab}}$.
10:            Update parametric experts parameters.
11:         **end for**
12:     **end for**
13: **else**
14:     {**Stage 2: Refine Semantic Guidance**}
15:     Freeze MoE-Adapter parameters.
16:     **for** local epoch $e = 1, ..., E$ **do**
17:         **for** batch $(x, y)$ from $D_c^t$ **do**
18:             Compute logits using Dual-Stream Cross-Attention.
19:             Compute loss $\mathcal{L} = \mathcal{L}_{\text{CE}}$.
20:             Update semantic-experts-related parameters.
21:         **end for**
22:     **end for**
23: **end if**
24: Let $\mathcal{L}_{\text{final}}$ be the average loss from the last local epoch.
25: Send updated $\boldsymbol{E}^{\text{shared}}$, $\mathcal{L}_{\text{final}}$, and $\boldsymbol{f}_c$ to server.

---

we create scenarios with 5 tasks (40 classes/task) and 10 tasks (20 classes/task). The class order is fixed for reproducibility.

- **Data Heterogeneity:** To simulate federated environments, we partition data among clients using a Dirichlet distribution over class labels, controlled by $\beta$. We test under moderate ($\beta = 0.5$), severe ($\beta = 0.1$), and IID settings.

**Domain-Incremental Learning on DomainNet.** This challenging scenario evaluates the model's ability to adapt to severe distribution shifts, where each task is a new visual domain.

- **Task Construction:** We use all six domains from the dataset (Clipart, Infograph, Painting, Quickdraw, Real, and Sketch) as a sequence of six tasks. The model learns these domains in alphabetical order.

- **Evaluation Protocol:** This scenario uses a specific protocol to measure cumulative knowledge. After training on each new domain, the model is evaluated on a dynamically expanding test set comprising the test splits of all previously learned domains plus the current one (e.g., after training on *Sketch*, the model is tested on the combined test sets of *Real* and *Sketch*). The evaluation uses a zero-shot classification head spanning all 345 classes of DomainNet.

**Multi-Domain Task-Incremental Learning on Fine-grained Datasets.** To further validate the generalizability of our framework, we construct a task-incremental scenario where the model must learn a sequence of entirely different datasets, assuming the task identity is known at inference time.

- **Task Construction:** We use a sequence of five distinct, fine-grained visual classification datasets, learned in the following order: Flowers102 → OxfordPets → Food101 → Caltech101 → DTD. Each dataset constitutes a single, separate task, testing the model's ability to acquire and retain diverse knowledge without interference.

### A.3 FORMAL DEFINITIONS OF METRICS AND LOSSES

#### A.3.1 EVALUATION METRICS

Here, we provide the formal definitions for the standard continual learning metrics used to evaluate our framework. Let $T$ be the total number of tasks, and let $a_{i,j}$ be the accuracy of the model on task $j$ after it has been trained on task $i$.

**Average Accuracy (Avg Acc)**   This metric measures the overall performance across all tasks after training is complete.

$$\text{Avg Acc} = \frac{1}{T} \sum_{j=1}^{T} a_{T,j} \tag{6}$$

**Forgetting Measure (Forget)**   This metric quantifies how much the model forgets about past tasks.

$$\text{Forget} = \frac{1}{T-1} \sum_{j=1}^{T-1} \left( \max_{k<T} a_{k,j} - a_{T,j} \right) \tag{7}$$

**Stability**   This measures the ability to retain knowledge of past tasks. After training on task $i$, it is the average accuracy on all prior tasks:

$$\text{Stability}_i = \frac{1}{i-1} \sum_{j=1}^{i-1} a_{i,j} \tag{8}$$

**Plasticity**   This measures the ability to learn the current new task. After training on task $i$, it is the accuracy on that task:

$$\text{Plasticity}_i = a_{i,i} \tag{9}$$

**Continual Utility**   This metric provides a score to evaluate the trade-off between stability and plasticity, defined as their weighted linear combination. The hyperparameter $\zeta \in [0,1]$ controls the balance, assigning a weight of $\zeta$ to stability and $(1 - \zeta)$ to plasticity. A larger $\zeta$ places more emphasis on stability.

$$\text{Utility}_i = \zeta \cdot \text{Stability}_i + (1 - \zeta) \cdot \text{Plasticity}_i \tag{10}$$

**Retrieval Recall@K**   This measures the percentage of queries where the correct match appears in the top-$K$ retrieved results. We evaluate both Image-to-Text (I2T) and Text-to-Image (T2I) retrieval at $K \in \{1, 5, 10\}$. For a query set of size $M$:

$$\text{Recall@K} = \frac{|\{i : \text{rank}(i) \leq K\}|}{M} \tag{11}$$

where $\text{rank}(i)$ denotes the rank of the correct match for query $i$. Higher recall indicates better retrieval capability.

**Alignment Score**   This quantifies the separation between matched and mismatched image-text pairs in the embedding space:

$$\text{Alignment Score} = \frac{1}{N} \sum_{i=1}^{N} s(v_i, t_i) - \frac{1}{N(N-1)} \sum_{i=1}^{N} \sum_{j \neq i} s(v_i, t_j) \tag{12}$$

where $v_i$ and $t_i$ denote the visual and textual embeddings of the $i$-th matched pair, $s(\cdot, \cdot)$ represents cosine similarity, and $N$ is the number of samples. A higher score indicates better cross-modal alignment.

**Peak Signal-to-Noise Ratio (PSNR)**    This metric is used to quantify the reconstruction quality in privacy attacks. It is defined via the Mean Squared Error (MSE) between the original image $I$ and the reconstructed image $\hat{I}$:

$$\text{PSNR} = 10 \cdot \log_{10}\left(\frac{\text{MAX}_I^2}{\text{MSE}}\right), \quad \text{where MSE} = \frac{1}{N}\sum_{i=1}^{N}(I_i - \hat{I}_i)^2 \tag{13}$$

Where $\text{MAX}_I$ is the maximum possible pixel value of the image, and $N$ is the total number of pixels. A lower PSNR indicates higher distortion, signifying better privacy protection.

**Structural Similarity Index (SSIM)**    This metric measures the perceptual structural similarity between two images, considering luminance, contrast, and structure. For an original image $x$ and a reconstructed image $y$:

$$\text{SSIM}(x, y) = \frac{(2\mu_x\mu_y + C_1)(2\sigma_{xy} + C_2)}{(\mu_x^2 + \mu_y^2 + C_1)(\sigma_x^2 + \sigma_y^2 + C_2)} \tag{14}$$

where $\mu_x, \mu_y$ represent the pixel sample means, $\sigma_x^2, \sigma_y^2$ are the variances, $\sigma_{xy}$ is the covariance, and $C_1, C_2$ are small constants for numerical stability. An SSIM value close to 0 implies no structural similarity, indicating successful defense against reconstruction.

### A.3.2    DETAILS OF PRIMARY AND REGULARIZATION LOSSES

**Cross-Entropy Loss ($\mathcal{L}_{CE}$)**    The primary objective for the main classification task is the standard cross-entropy loss. It is designed to minimize the dissimilarity between the model's predicted probability distribution and the ground-truth label distribution. The formula is:

$$\mathcal{L}_{CE} = -\sum_{i=1}^{C} y_i \log(\hat{y}_i), \tag{15}$$

where $C$ is the total number of classes, $\boldsymbol{y}$ is the one-hot encoded ground-truth label vector, and $\hat{\boldsymbol{y}}$ is the predicted probability distribution produced by applying the Softmax function to the model's final logits. Minimizing this loss guides the model to assign the highest possible probability to the correct class.

**MoE Load Balancing Loss ($\mathcal{L}_{\mathbf{moe}}$)**    To prevent the MoE router from consistently favoring a small subset of experts (a phenomenon known as expert collapse), we employ an auxiliary load-balancing loss. This loss encourages the router to distribute the computational load as evenly as possible across all available experts, ensuring that all experts are sufficiently trained. Following common practice, we define this loss as:

$$\mathcal{L}_{\text{moe}} = \alpha \cdot N \sum_{i=1}^{N} f_i \cdot P_i, \tag{16}$$

where $N$ is the number of experts and $\alpha$ is a scaling hyperparameter. For a batch of tokens, $f_i$ is the fraction of tokens dispatched to expert $i$, and $P_i$ is the average router probability (the gating value) assigned to expert $i$ over the batch. This objective incentivizes the router to assign higher probabilities ($P_i$) to experts that are utilized less frequently (low $f_i$), thus promoting balanced expert utilization throughout training.

### A.4    IMPLEMENTATION AND ARCHITECTURAL DETAILS

### A.4.1    DUAL-STREAM CROSS-ATTENTION MECHANISM

To create a dynamic, content-aware fusion of knowledge that is native to the CLIP architecture, we introduce a Dual-Stream Cross-Attention mechanism. This design moves beyond static feature combination by using the visual feature of an input image, $\boldsymbol{F}_{\text{img}}$, as a dynamic Query to actively interrogate the available textual knowledge. This process unfolds in two parallel streams.

**Local Knowledge Stream.** First, to capture client-specific nuances, each client learns a private, local prompt $\boldsymbol{p}_c^{\text{local}}$. This prompt is prepended to the class token embeddings and processed by the text encoder to produce the local text feature set, $\boldsymbol{z}_c^{\text{local}}$. The local knowledge stream then attends to this private local prompt feature, which serves as both Key and Value. This produces an image-aware local representation and its corresponding expert logits:

$$\text{logits}_{\text{local}} = \text{Cross-Attention}(\boldsymbol{F}_{\text{img}}, \boldsymbol{z}_c^{\text{local}}, \boldsymbol{z}_c^{\text{local}}) \tag{17}$$

**Shared Knowledge Stream.** Concurrently, to evaluate the relevance of the generalized knowledge from the federation, the shared knowledge stream attends the same image query to the set of $k$ received shared prompt features, $\boldsymbol{Z}_s$. This computes the shared logits:

$$\text{logits}_{\text{shared}} = \text{Cross-Attention}(\boldsymbol{F}_{\text{img}}, \boldsymbol{Z}_s, \boldsymbol{Z}_s) \tag{18}$$

**Final Fusion.** Finally, to create a prediction that flexibly balances local expertise with global consensus, the logits from both streams are dynamically fused via a weighted sum, controlled by a hyperparameter $\lambda$:

$$\text{Logits}_{\text{final}} = \lambda \cdot \text{logits}_{\text{local}} + (1 - \lambda) \cdot \text{logits}_{\text{shared}} \tag{19}$$

This mechanism allows the model to dynamically weigh the importance of local versus global semantic guidance based on the task and data.

### A.4.2 EXPERIMENTAL DETAILS FOR PRIVACY ANALYSIS

In this section, we detail the implementation of the noise injection mechanism used for robustness evaluation and the setup for the gradient-based reconstruction attacks presented in the main text (Sec. 4.2).

**Noise Injection Mechanism.** To evaluate the system's resilience under strict Differential Privacy (DP) constraints, we adopt the noise injection protocol established in MoAFCL. Specifically, Laplacian noise is injected into the aggregated client feature summary $\tilde{\boldsymbol{f}}_c \in \mathbb{R}^{512}$ before it serves as input to the server-side gating mechanism. The perturbed feature vector, $\tilde{\boldsymbol{f}}_c^{\text{dp}}$, is computed as follows:

$$\tilde{\boldsymbol{f}}_c^{\text{dp}} = \tilde{\boldsymbol{f}}_c + \mathbf{n}, \quad \text{where } n_j \sim \text{Lap}(0, \sigma) \text{ for } j = 1, \ldots, 512, \tag{20}$$

where $\sigma$ is the noise multiplier controlling the scale of the perturbation. This setup allows us to simulate varying degrees of privacy budgets and assess the trade-off between privacy protection and the utility of the gating mechanism.

**Reconstruction Attack Setup.** To empirically verify the defense capability against visual leakage, we conduct gradient-based feature inversion attacks. The objective is to reconstruct the original input image $\boldsymbol{x}$ solely from its compressed feature representation. We treat this as an optimization problem where a dummy image $\hat{\boldsymbol{x}}$ is iteratively updated to match the target features. Specifically, we employ the Adam optimizer for $1,000$ steps to ensure convergence. The optimization minimizes a composite objective function $\mathcal{L}_{\text{attack}}$ consisting of three terms: Cosine Similarity loss to align the feature direction, Total Variation (TV) regularization to encourage spatial smoothness in the reconstructed image, and L2 regularization to penalize pixel outliers.

## A.5 ADDITIONAL RESULTS AND VISUALIZATIONS

To further substantiate the claims made in the main paper, this section provides additional visualizations and analyses of our framework's performance.

### A.5.1 DETAILED PERFORMANCE TRAJECTORY ANALYSIS

Figure 5 presents a comprehensive visualization of the accuracy trajectory for Fed-Duet against baseline methods across Non-IID scenarios. These plots complement the summary results in Table 1 of the main paper by showing the performance degradation as more tasks are learned. Across all settings, our method consistently establishes the highest accuracy. More importantly, its performance curve exhibits a significantly flatter slope compared to all baselines, visually confirming its superior resistance to catastrophic forgetting, especially in the Non-IID scenarios.

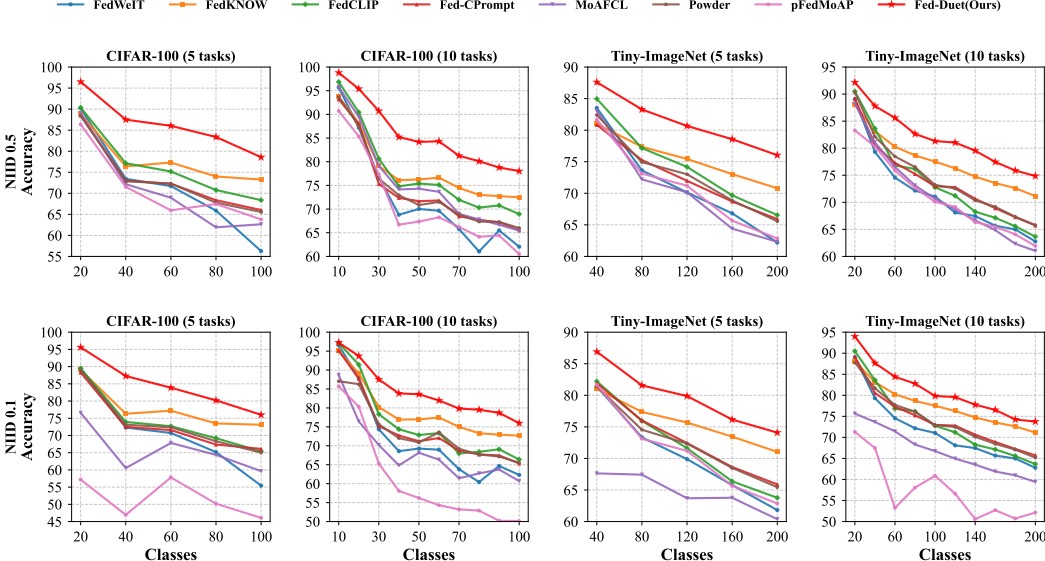

Figure 5: Detailed performance trajectories on CIFAR-100 and Tiny-ImageNet. **Rows**: The top row displays results under the IID setting, while the bottom row shows the more challenging Non-IID ($\beta = 0.1$) setting. **Columns**: From left to right, the columns correspond to CIFAR-100 (5 tasks), CIFAR-100 (10 tasks), Tiny-ImageNet (5 tasks), and Tiny-ImageNet (10 tasks).

Table 8: Multi-domain Task-incremental learning (MTIL) performance on five benchmark datasets. The final columns show the average accuracy and the $\Delta$ (blue for lower) compared to our method.

| Method | Flowers102 | OxfordPets | Food101 | Caltech101 | DTD | Average | $\Delta$ |
|---|---|---|---|---|---|---|---|
| FedWeIT | 62.03 | 53.97 | 66.35 | 69.50 | 71.68 | 64.71 | -8.61 |
| FedKNOW | 63.69 | 53.98 | 64.81 | 70.02 | 72.31 | 64.96 | -8.36 |
| FedCLIP | 68.97 | 64.67 | 71.83 | 76.31 | 79.84 | 72.32 | -1.00 |
| Fed-CPrompt | 63.96 | 54.10 | 64.95 | 70.21 | 72.46 | 65.14 | -8.18 |
| Powder | 65.48 | 58.37 | 67.87 | 72.32 | 76.47 | 68.10 | -5.22 |
| pFedMoAP | 66.56 | 59.58 | 69.82 | 73.22 | 77.77 | 69.39 | -3.93 |
| MoAFCL | 63.18 | 54.09 | 66.32 | 71.45 | 74.85 | 65.98 | -7.34 |
| Fed-Duet (Ours) | **70.35** | **64.97** | **73.33** | **77.77** | **80.17** | **73.32** | – |

### A.5.2 MULTI-DOMAIN TASK-INCREMENTAL LEARNING

In the main body of our paper, we extensively evaluated our framework under class-incremental and domain-incremental learning scenarios. To further demonstrate the robustness and transferability of our proposed Fed-Duet framework, we present an auxiliary experiment in this section under the **Multi-Domain Task-Incremental Learning** setting.

**Analysis of Multi-Domain Task-Incremental.** The performance of all methods on this Multi-domain Task-incremental benchmark is presented in Table 8. The results unequivocally show that our Fed-Duet framework achieves the highest accuracy across all five datasets in the sequence. This consistent state-of-the-art performance in a third, distinct continual learning paradigm provides strong evidence for the generalizability of our approach. It confirms that the superiority of Fed-Duet is not confined to a specific problem formulation but is instead a robust attribute stemming from its core design, further validating its advanced capabilities in diverse continual learning environments.

### A.5.3 AUXILIARY EXPERIMENT: PERFORMANCE WITH A COMPRESSED BACKBONE

In our main experiments, we utilized the full CLIP ViT-B/16 model to ensure a fair comparison with prior works. However, practical federated learning scenarios often involve clients with **limited**

Table 9: Performance comparison on CIFAR-100 and Tiny-ImageNet under compression strategy.

(a) CIFAR-100

| Method | T=5 | | T=10 | |
|---|---|---|---|---|
| | Avg ↑ | Last ↑ | Avg ↑ | Last ↑ |
| FedWeIT | 38.71 | 25.21 | 39.81 | 24.52 |
| FedKNOW | 42.08 | 34.60 | 46.69 | 35.86 |
| FedCLIP | 49.39 | 34.27 | 49.39 | 32.87 |
| Fed-CPrompt | 34.75 | 24.10 | 37.69 | 23.88 |
| Powder | 34.65 | 23.71 | 37.51 | 23.55 |
| pFedMoAP | 37.99 | 25.28 | 39.73 | 25.15 |
| MoAFCL | 40.69 | 27.47 | 42.29 | 27.17 |
| Fed-Duet (Ours) | 61.31 | 49.04 | 63.29 | 46.45 |

(b) Tiny-ImageNet

| Method | T=5 | | T=10 | |
|---|---|---|---|---|
| | Avg ↑ | Last ↑ | Avg ↑ | Last ↑ |
| FedWeIT | 23.67 | 16.90 | 25.64 | 15.58 |
| FedKNOW | 28.60 | 22.86 | 32.01 | 24.02 |
| FedCLIP | 35.15 | 22.82 | 35.25 | 21.18 |
| Fed-CPrompt | 23.80 | 15.98 | 25.36 | 15.86 |
| Powder | 24.03 | 15.86 | 25.46 | 15.82 |
| pFedMoAP | 27.75 | 18.26 | 28.63 | 17.29 |
| MoAFCL | 24.81 | 16.91 | 25.81 | 15.68 |
| Fed-Duet (Ours) | 46.62 | 34.96 | 47.58 | 33.29 |

**computational resources**, necessitating more lightweight models. To evaluate the robustness and transferability of our framework under such constraints, we conduct an auxiliary experiment where all methods are built upon a compressed CLIP backbone.

**Pruning the CLIP Backbone.** We adopt a simple yet effective layer-dropping strategy to compress the CLIP backbone. Specifically, we prune the model by removing the first two and last two transformer layers from both the vision and text encoders. While more sophisticated pruning techniques could be explored in future work, this method serves to effectively reduce the model's computational footprint for this analysis.

**Sustained Performance with a Compressed Model.** As shown in Table 9, Fed-Duet maintains its state-of-the-art performance even with the compressed backbone, consistently outperforming all baselines. This result demonstrates that the superiority of our framework is intrinsic to its architectural design, not merely a byproduct of a large-scale model. Retaining this performance advantage validates the robustness and practical applicability of Fed-Duet in resource-constrained federated environments.

Table 10: Hyperparameter analysis for Fed-Duet on CIFAR-100.

(a) Impact of Experts (E) and Top-K (k).

| Experts (E) | Top-K (k) | | |
|---|---|---|---|
| | k=2 | k=4 | k=8 |
| 2 | 84.97 | – | – |
| 4 | 85.77 | – | – |
| 8 | 86.21 | 86.42 | – |
| 16 | 87.12 | 86.98 | 86.75 |

(b) Impact of K-means clusters.

| Clusters | Accuracy (%) |
|---|---|
| 8 | 86.23 |
| 16 | 86.17 |
| 32 | 86.10 |
| 64 | 86.21 |
| 128 | 86.34 |

Table 11: Sensitivity and Ablation Analysis of FedDuet.

(a) Loss Function Weights ($\alpha, \eta, \gamma$)

| Value | $\alpha$ | $\eta$ | $\gamma$ |
|---|---|---|---|
| 0.01 | 86.25 | 86.16 | 86.20 |
| 0.1 | 86.23 | 86.14 | 86.19 |
| 1.0 | 86.19 | 86.15 | 86.17 |
| 10.0 | 85.99 | 86.19 | 86.18 |

(b) Fusion Coefficient $\lambda$

| $\lambda$ | CIFAR-100 (%) | Tiny-ImageNet (%) |
|---|---|---|
| 0.00 | 31.05 | 25.68 |
| 0.25 | 85.71 | 82.06 |
| 0.50 | 86.21 | 82.60 |
| 0.75 | 85.85 | 82.50 |
| 1.00 | 85.71 | 82.46 |

A.5.4 HYPERPARAMETER SENSITIVITY AND ABLATION

We perform a comprehensive study on FedDuet's hyperparameters, including the number of Experts, Top-K selection, K-means clusters, loss function weights ($\alpha, \eta, \gamma$), and the fusion coefficient $\lambda$.

As shown in Table 10, the model exhibits strong robustness to variations in Experts, Top-K, and K-means clusters. Table 11 further demonstrates that FedDuet maintains stable performance across a wide range of loss weight values, while moderate fusion ($\lambda = 0.50$) achieves optimal results. Overall, these analyses highlight the method's low sensitivity to hyperparameters, making it practical for deployment without extensive tuning.

### A.5.5   ABLATION STUDIES

Table 12: Ablation study on our Progressive Decoupled Optimization strategy. Best results are in **bold**. The blue values ($\Delta$) indicate a score lower than our method.

| Training Strategy | CIFAR-100 | Tiny-ImageNet | Average | $\Delta$ |
|---|---|---|---|---|
| Semantic Experts First | 85.53 | 81.28 | 83.41 | -0.98 |
| Joint Training | 86.02 | 81.88 | 83.95 | -0.44 |
| **Parametric Experts First (Ours)** | **86.21** | **82.56** | **84.39** | – |

**Optimization Strategy.**   We evaluate different training strategies to validate the effectiveness of our Progressive Decoupled Optimization. Prioritizing parametric experts consistently yields the best performance, confirming that separating parametric and semantic training stages facilitates more effective knowledge acquisition and representation learning compared to joint or semantic-first training.

Table 13: Impact of Stage Transition Timing on Performance. R denotes the total communication rounds. Best results are in **bold**. The blue values ($\Delta$) indicate a score lower than our method.

| Transition Timing | CIFAR-100 | | Tiny-ImageNet | | Average | $\Delta$ |
|---|---|---|---|---|---|---|
| | IID | Non-IID ($\beta$=0.1) | IID | Non-IID ($\beta$=0.1) | | |
| R/4 | 84.86 | 82.86 | 81.29 | 78.73 | 81.94 | -1.53 |
| 3R/4 | 86.12 | 84.29 | 82.88 | 80.41 | 83.43 | -0.04 |
| **R/2 (Ours)** | **86.21** | **84.58** | **82.60** | **80.48** | **83.47** | – |

**Stage Transition Timing.**   We investigate the impact of stage transition timing within Progressive Decoupled Optimization. Transitioning at the midpoint of training strikes the best balance, providing parametric experts sufficient time to develop foundational representations before semantic experts refine them. Moving the transition earlier compromises learning stability, while delaying it offers little additional benefit, highlighting the importance of timely stage scheduling.

Table 14: Comparison of Prompt Pool Initialization Strategies on CIFAR-100. Best results are in **bold**. The blue values ($\Delta$) indicate a score lower than K-means initialization.

| Initialization Strategy | IID | Non-IID ($\beta$=0.5) | Non-IID ($\beta$=0.1) | Average | $\Delta$ |
|---|---|---|---|---|---|
| Random | 85.93 | 83.88 | 84.02 | 84.61 | -0.58 |
| Uniform | 85.96 | 84.23 | 84.09 | 84.76 | -0.43 |
| **K-means (Ours)** | **86.21** | **84.77** | **84.58** | **85.19** | – |

**Prompt Pool Initialization.**   We analyze different strategies for initializing the prompt pool. K-means clustering consistently produces more robust and generalizable prompts, particularly under heterogeneous data distributions. This demonstrates that a well-structured initial prompt set enhances the model's ability to handle diverse client data.

**Prompt Selection Strategies.**   We compare dynamic prompt selection via our Gating Network against random assignment. The Gating Network consistently improves performance across all settings, especially in heterogeneous federated environments. This highlights the advantage of adaptive selection in leveraging client-specific features and maximizing the utility of the prompt pool.

Table 15: Ablation study on prompt selection strategies. We compare our Gating Network with random selection of top-K prompts across different datasets and data distributions. The $\Delta$ indicates the performance improvement relative to random selection, highlighted in blue.

| Selection Strategies | CIFAR-100 | | Tiny-ImageNet | | Average | $\Delta$ |
|---|---|---|---|---|---|---|
| | IID | Non-IID ($\beta$=0.1) | IID | Non-IID ($\beta$=0.1) | | |
| Random Selection | 85.96 | 83.27 | 82.07 | 79.58 | 82.22 | -1.25 |
| **Gating Network (Ours)** | **86.21** | **84.58** | **82.60** | **80.48** | **83.47** | – |

Table 16: Robustness Analysis on CIFAR-100 and Tiny-ImageNet. We extend the main results by reporting mean accuracy $\pm$ standard deviation across multiple runs to demonstrate the robustness and reproducibility of our approach.

| Method | CIFAR-100 | | Tiny-ImageNet | |
|---|---|---|---|---|
| | IID | Non-IID ($\beta$=0.1) | IID | Non-IID ($\beta$=0.1) |
| FedWeIT | $72.52 \pm 0.62$ | $70.43 \pm 0.64$ | $72.06 \pm 0.67$ | $71.06 \pm 0.64$ |
| FedKNOW | $78.47 \pm 0.52$ | $77.25 \pm 0.43$ | $76.24 \pm 0.73$ | $74.61 \pm 0.84$ |
| FedCLIP | $78.21 \pm 0.37$ | $74.17 \pm 0.57$ | $76.14 \pm 0.51$ | $71.94 \pm 0.57$ |
| Fed-CPrompt | $73.41 \pm 0.42$ | $73.00 \pm 0.33$ | $73.09 \pm 0.45$ | $72.48 \pm 0.32$ |
| Powder | $73.32 \pm 0.34$ | $72.71 \pm 0.51$ | $72.98 \pm 0.41$ | $72.71 \pm 0.62$ |
| pFedMoAP | $76.61 \pm 0.23$ | $52.63 \pm 1.02$ | $74.47 \pm 0.20$ | $55.47 \pm 1.23$ |
| MoAFCL | $77.93 \pm 0.13$ | $65.96 \pm 1.23$ | $74.16 \pm 0.59$ | $62.51 \pm 1.17$ |
| **Fed-Duet (Ours)** | $\mathbf{86.21 \pm 0.11}$ | $\mathbf{84.58 \pm 0.19}$ | $\mathbf{82.60 \pm 0.13}$ | $\mathbf{80.48 \pm 0.21}$ |

### A.5.6 ROBUSTNESS ANALYSIS WITH STANDARD DEVIATION

Table 16 extends the main results by reporting mean accuracy with standard deviation across multiple runs. FedDuet achieves both the highest accuracy and lowest standard deviation across all settings. For example, on CIFAR-100 Non-IID ($\beta$=0.1), FedDuet obtains $84.58 \pm 0.19$ compared to the second-best FedKNOW at $77.25 \pm 0.43$, demonstrating superior stability and reproducibility.

### A.5.7 CROSS-MODAL ALIGNMENT AND RETRIEVAL PERFORMANCE

We conduct comprehensive evaluations to verify that **FedDuet effectively preserves and enhances cross-modal alignment** in federated continual learning scenarios. Our analysis covers three perspectives: alignment scores across tasks, retrieval performance after cumulative training, and alignment evolution during two-stage training.

Table 17: Cross-Modal Retrieval Performance on Tiny-ImageNet. We evaluate all methods after completing the final task (Task 4) on the cumulative test set. Metrics include Image-to-Text (I2T) and Text-to-Image (T2I) Recall@K. Results are averaged across 5 clients. Best results are in **bold**. The blue values ($\Delta$) indicate a score lower than our method. Results are averaged across 5 clients.

| Method | Image-to-Text (I2T) | | | Text-to-Image (T2I) | | | Average | $\Delta$ |
|---|---|---|---|---|---|---|---|---|
| | R@1 $\uparrow$ | R@5 $\uparrow$ | R@10 $\uparrow$ | R@1 $\uparrow$ | R@5 $\uparrow$ | R@10 $\uparrow$ | | |
| FedCLIP | 64.38 | 85.50 | 90.61 | 88.10 | 98.30 | 99.50 | 87.73 | -4.19 |
| FedWeIT | 63.65 | 85.89 | 90.99 | 87.90 | 98.40 | 99.50 | 87.72 | -4.20 |
| FedCPrompt | 63.83 | 85.89 | 90.61 | 88.00 | 98.30 | 99.80 | 87.74 | -4.18 |
| Powder | 63.85 | 85.48 | 90.66 | 88.90 | 98.80 | 99.60 | 87.88 | -4.04 |
| pFedMoAP | 63.52 | 85.71 | 90.93 | 87.60 | 98.40 | 99.40 | 87.59 | -4.33 |
| FedKNOW | 63.65 | 85.89 | 90.99 | 87.90 | 98.40 | 99.50 | 87.72 | -4.20 |
| MoAFCL | 63.15 | 85.79 | 90.55 | 86.90 | 98.30 | 99.30 | 87.33 | -4.59 |
| FedDuet (Ours) | **74.67** | **92.46** | **95.91** | **92.90** | **99.60** | **100.0** | **91.92** | – |

**Cross-Modal Retrieval Performance.** Tables 17 (and the corresponding CIFAR-100 Table 3) report Image-to-Text (I2T) and Text-to-Image (T2I) retrieval metrics on cumulative test sets. **FedDuet outperforms all baselines across both datasets**, with the largest gains observed in top-1 retrieval. This improvement highlights that preserving cross-modal alignment directly translates into more accurate retrieval, demonstrating that our approach maintains the fundamental vision-language understanding of CLIP models beyond mere classification accuracy.

Table 18: Cross-Modal Alignment Score Across Tasks on Tiny-ImageNet. We track the alignment score after each task to demonstrate the ability to maintain cross-modal alignment during continual learning. Results are averaged across 5 clients. Best results are in **bold**. The blue values ($\Delta$) indicate a score lower than our method. Results are averaged across 5 clients.

| Method | Task 0 | Task 1 | Task 2 | Task 3 | Task 4 | Average | $\Delta$ |
|---|---|---|---|---|---|---|---|
| FedCLIP | 0.0821 | 0.0829 | 0.0837 | 0.0835 | 0.0833 | 0.0831 | -0.1229 |
| FedWeIT | 0.0823 | 0.0830 | 0.0838 | 0.0835 | 0.0834 | 0.0832 | -0.1228 |
| FedCPrompt | 0.0824 | 0.0831 | 0.0837 | 0.0834 | 0.0833 | 0.0832 | -0.1228 |
| Powder | 0.0823 | 0.0830 | 0.0836 | 0.0835 | 0.0832 | 0.0831 | -0.1229 |
| pFedMoAP | 0.0820 | 0.0829 | 0.0839 | 0.0834 | 0.0832 | 0.0831 | -0.1229 |
| FedKNOW | 0.0823 | 0.0830 | 0.0838 | 0.0835 | 0.0834 | 0.0832 | -0.1228 |
| MoAFCL | 0.0824 | 0.0828 | 0.0835 | 0.0833 | 0.0831 | 0.0830 | -0.1230 |
| FedDuet (Ours) | **0.1900** | **0.1962** | **0.2065** | **0.1980** | **0.2061** | **0.2060** | – |

**Alignment Across Tasks.** Tables 18 (and the corresponding CIFAR-100 Table 4) track cross-modal alignment scores after each task. **FedDuet consistently achieves the highest alignment scores**, substantially surpassing all baseline methods. While conventional approaches maintain relatively low and stable alignment, FedDuet demonstrates significantly stronger alignment that is stable across sequential tasks, indicating its ability to maintain robust image-text associations throughout continual learning.

Table 19: Cross-Modal Alignment Score Evolution Across Training Rounds. Stage 1 (Rounds 1–5) corresponds to parametric expert training, and Stage 2 (Rounds 6–10) corresponds to semantic prompt training.

| Round | Stage | Alignment Score | Positive Sim | Negative Sim |
|---|---|---|---|---|
| 1 | Stage 1 | 0.1496 | 0.2851 | 0.1355 |
| 2 | Stage 1 | 0.1664 | 0.2932 | 0.1267 |
| 3 | Stage 1 | 0.1775 | 0.2942 | 0.1166 |
| 4 | Stage 1 | 0.1873 | 0.2945 | 0.1072 |
| 5 | Stage 1 | 0.1911 | 0.2965 | 0.1054 |
| **Stage 1 Change** | | **+0.0415** | **+0.0114** | **-0.0301** |
| 6 | Stage 2 | 0.1955 | 0.2981 | 0.1027 |
| 7 | Stage 2 | 0.1953 | 0.2979 | 0.1026 |
| 8 | Stage 2 | 0.1952 | 0.2979 | 0.1027 |
| 9 | Stage 2 | 0.1952 | 0.2979 | 0.1027 |
| 10 | Stage 2 | 0.1954 | 0.2980 | 0.1027 |
| **Stage 2 Change** | | **-0.0001** | **-0.0001** | **+0.0000** |
| **Overall Change (R1$\rightarrow$R10)** | | **+0.0458** | **+0.0130** | **-0.0328** |

**Alignment Dynamics Across Two-Stage Training.** FedDuet is designed to balance continual task adaptation with stable cross-modal grounding. To assess how our two-stage training supports this goal, we track alignment scores and cross-modal retrieval performance on Task 0 throughout training (Tables 19 and 20).

**Stage 1 (parametric expert training)** progressively strengthens cross-modal grounding, reflected by consistent improvements in alignment scores and I2T retrieval quality.

Table 20: Cross-Modal Retrieval Performance Evolution Across Training Rounds. We report Image-to-Text (I2T) and Text-to-Image (T2I) Recall@K metrics. Stage 1 corresponds to parametric expert training (Rounds 1–5), Stage 2 corresponds to semantic prompt training (Rounds 6–10).

| Round | Stage | I2T Recall (%) | | | T2I Recall (%) | | |
|---|---|---|---|---|---|---|---|
| | | R@1 | R@5 | R@10 | R@1 | R@5 | R@10 |
| 1 | Stage 1 | 94.66 | 99.75 | 99.95 | 100.00 | 100.00 | 100.00 |
| 2 | Stage 1 | 95.54 | 99.82 | 99.94 | 100.00 | 100.00 | 100.00 |
| 3 | Stage 1 | 96.00 | 99.86 | 99.95 | 100.00 | 100.00 | 100.00 |
| 4 | Stage 1 | 96.33 | 99.82 | 99.96 | 100.00 | 100.00 | 100.00 |
| 5 | Stage 1 | 96.38 | 99.88 | 99.96 | 99.00 | 100.00 | 100.00 |
| **Stage 1 Change** | | **+1.72** | **+0.13** | **+0.01** | **-1.00** | **0.00** | **0.00** |
| 6 | Stage 2 | 96.68 | 99.85 | 99.96 | 100.00 | 100.00 | 100.00 |
| 7 | Stage 2 | 96.44 | 99.86 | 99.94 | 100.00 | 100.00 | 100.00 |
| 8 | Stage 2 | 96.68 | 99.88 | 99.97 | 99.00 | 100.00 | 100.00 |
| 9 | Stage 2 | 96.69 | 99.88 | 99.96 | 100.00 | 100.00 | 100.00 |
| 10 | Stage 2 | 96.51 | 99.86 | 99.98 | 100.00 | 100.00 | 100.00 |
| **Stage 2 Change** | | **-0.17** | **+0.01** | **+0.02** | **+1.00** | **0.00** | **0.00** |
| **Overall Change (R1→R10)** | | **+1.85** | **+0.11** | **+0.03** | **0.00** | **0.00** | **0.00** |

**Stage 2 (semantic prompt training)** preserves this grounding: with experts frozen, semantic prompts adapt while alignment scores and retrieval metrics remain stable. This indicates that the learned cross-modal structure acts as a reliable anchor during downstream prompt updates.

Overall, the results show that FedDuet's decoupled training cleanly separates representation grounding from semantic adaptation, enabling continual refinement without compromising cross-modal consistency.

### A.5.8 FEATURE SPACE VISUALIZATION VIA T-SNE

To qualitatively verify the efficacy of our framework in mitigating catastrophic forgetting, we employ t-SNE to visualize the learned feature space after the model has trained on all sequential tasks. As depicted in Figure 6, we project the feature embeddings of test samples into a 2D space. The visualization reveals that Fed-Duet maintains a highly structured and separable feature space, evidenced by the dense and distinct clusters formed by each task's samples. This clear spatial separation serves as strong qualitative evidence that our framework preserves the integrity of feature representations for earlier tasks, thereby effectively mitigating catastrophic forgetting at the feature level.

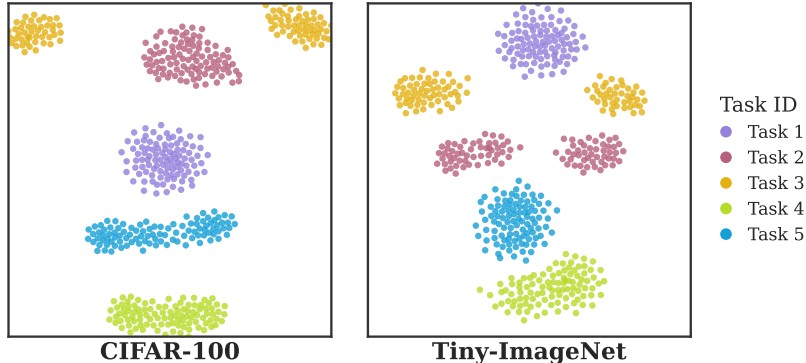

Figure 6: t-SNE visualization of feature representations for test samples from all tasks on CIFAR-100 (Left) and Tiny-ImageNet (Right) after the final task. Each color corresponds to a different continual learning task.

### A.5.9 ANALYSIS OF EXPERT UTILIZATION

The expert activation patterns, visualized in Figure 7, reveal both effective load balancing and a high degree of specialization. The heatmaps show a distributed activation strategy, confirming that our model avoids expert collapse by engaging a diverse set of experts. Furthermore, we observe a strong tendency for specific experts to be responsible for distinct task categories, particularly in the deeper layers. This dual observation of balance and specialization mutually corroborates the findings from our t-SNE visualization (Figure 6), which similarly indicates that experts form distinct, specialized clusters.

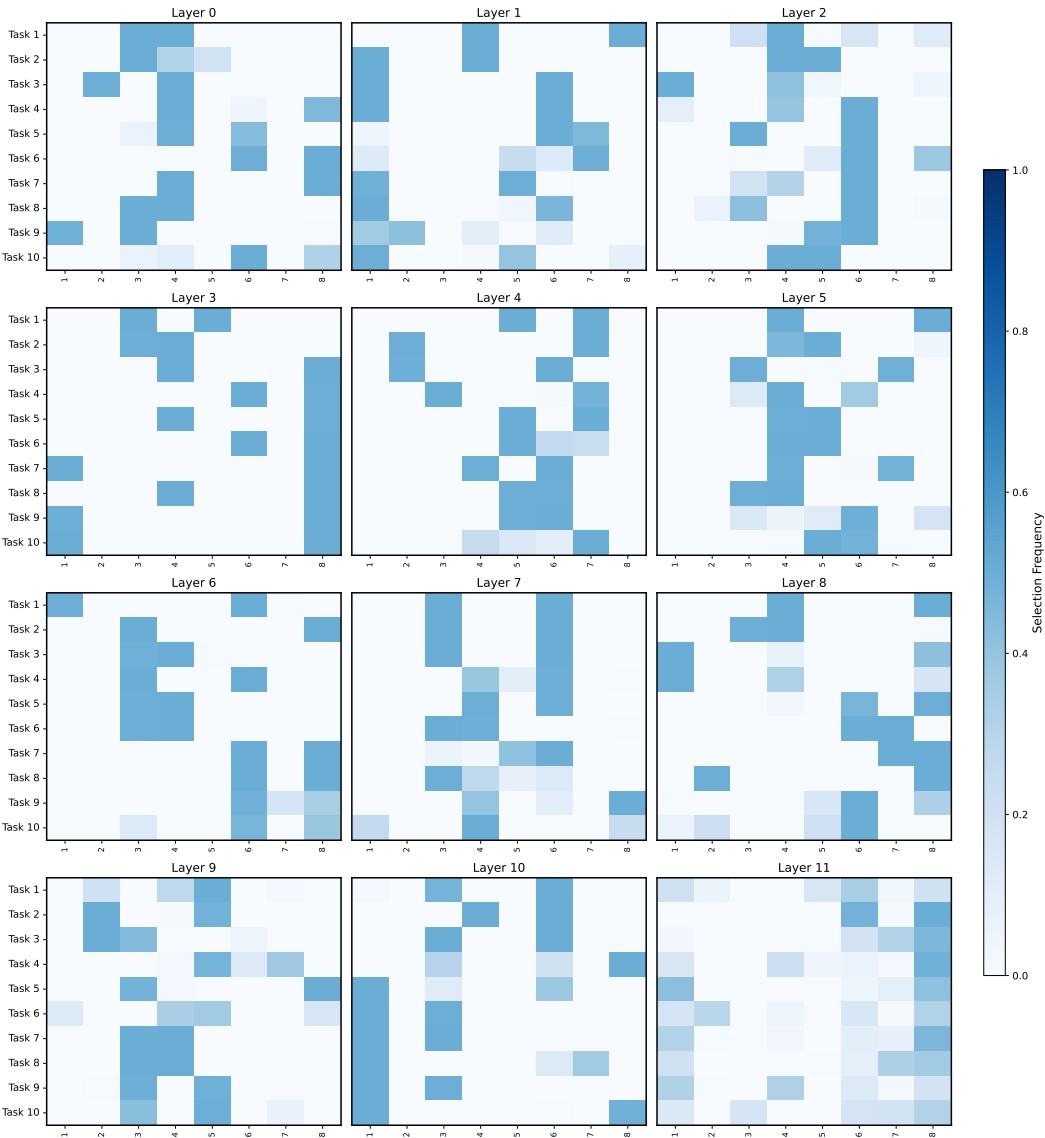

Figure 7: Visualization of expert selection frequency per task in our proposed FedDuet framework. The heatmaps illustrate which of the 8 experts are activated for each of the 10 sequential tasks across 12 transformer layers. The color intensity corresponds to the selection frequency, highlighting how expert utilization evolves for different tasks and layers.

