# OpenReview forum: "Fed-Duet: Dual Expert-Orchestrated Framework for Continual Federated Vision-Language Learning"
_ICLR.cc/2026/Conference — ICLR 2026 Poster_

### Official Review · Reviewer_9Jgs · 2025-10-15

**Soundness:** 3
**Presentation:** 2
**Contribution:** 3
**Rating:** 6
**Confidence:** 2

**Summary:**

This paper introduces Fed-Duet, a framework to address continual federated learning for CLIP models. The key idea is a dual-channel expert system: The paper adopts (1) semantic experts: server-coordinated prompts for high-level multimodal alignment and (2) parametric experts, i.e., client-specific adapters. A cross-attention fusion mechanism integrates these two channels, while auxiliary objectives preserve cross-modal consistency and mitigate catastrophic forgetting. Extensive experiments on multiple benchmarks demonstrate the promising performance of the method in both accuracy and efficiency.

**Strengths:**

1. The proposed method achieves promising results on multiple benchmarks.
2. The experiments are thorough and comprehensive.

**Weaknesses:**

1. The proposed method is complex, I would recommend refine/extend a bit on the details of the method design and different components. Also, I personally don't think "Semantic Experts" is a good naming for textual prompt.
2. There are too much information included in Figure 1, maybe considering separate it into multiple figures.
3. The authors mentioned about "privacy-preserving feature" of the client images will be uploaded to the server. However, there lacks details about how the strength of the noise and whether it is still possible to reproduce the images at server-side. The author should discuss this more.
4. How would the attention gating influence the model performance? There seems to be no ablation about this specific component.

**Questions:**

Please refer to the Weaknesses section.

---

> ### Author Response · Authors · 2025-11-20
> **Author Response to Reviewer 9Jgs (Part1)**
>
> > **[W1]** The proposed method is complex, I would recommend refine/extend a bit on the details of the method design and different components. Also, I personally don't think "Semantic Experts" is a good naming for textual prompt.
>
>
> We thank the reviewer for raising this concern. Although our full framework contains multiple components, the core idea of FedDuet is conceptually simple:
>
> **1. Simplifying the Framework**
>
> **"The server orchestrates transferable knowledge, while clients adapt via two complementary pathways."**
> * **Pathway 1 (Semantic):** Adjusts lightweight prompts to align input data with task semantics.
> * **Pathway 2 (Parametric):** Adapts local adapters to transform visual features for stability.
>
> **2. Justification for "Semantic Experts"**
>
> We respectfully argue that "Semantic Experts" is a deliberate and accurate functional descriptor. We use this term because these learnable vectors are specifically designed to **encapsulate and provide semantic knowledge**, rather than serving merely as static formatting tokens.
>
> The naming directly reflects their physical meaning in our framework:
> * **Local Semantic Expert:** The client maintains this module to represent **client-specific semantics** (e.g., local class attributes and task contexts).
> * **Shared Semantic Experts:** The server provides these modules to reflect **globally transferable semantics** (e.g., universal knowledge distilled from the prompt pool).
>
>
> **3. Functional Distinction**
>
> By dynamically fusing these Local and Shared sources via the Attention Gate, these experts act as **"Strategic Orchestrators."** They operate at the input level to provide top-down semantic guidance *before* the visual backbone (the Parametric Experts) executes feature transformation. This distinct role—managing and injecting semantic context—justifies the terminology "Semantic Experts."
>
> ---
>
> > **[W2]** There are too much information included in Figure 1, maybe considering separate it into multiple figures.
>
>
>
> We thank the reviewer for the helpful suggestion regarding the complexity of Figure 1. We fully agree that the original visualization was dense.
>
> To improve clarity and readability, we have **restructured Figure 1 into two distinct sub-figures** to decouple the global workflow from local adaptation details:
> * **Figure 1(a): FedDuet Training Overview.** This part illustrates the macro-level interaction between the server (Knowledge Orchestrator) and clients.
> * **Figure 1(b): Client Local Training.** This part zooms in on the specific dual-expert optimization process within a single client.
>
> This separation allows for a cleaner presentation of both the system-level architecture and the client-side mechanisms. The updated figures can be found in the revised manuscript.

---

> > ### Author Response · Authors · 2025-11-20
> > **Author Response to Reviewer 9Jgs (Part2)**
> >
> > > **[W3]** The authors mentioned about "privacy-preserving feature" of the client images will be uploaded to the server. However, there lacks details about how the strength of the noise and whether it is still possible to reproduce the images at server-side. The author should discuss this more.
> >
> > We thank the reviewer for raising this important question regarding the details of the privacy mechanism and reconstruction risks.
> >
> >
> > We have significantly improved the description in the revised manuscript. Specifically, we have expanded **Section 4.2 (Privacy Compatibility and Robustness Analysis) and Section 3.1 (Adaptive Dispatch Mechanism)** to explicitly provide the missing details:
> >
> > **Clarification of Role:**
> > We clarified that the noise injection is **not** the primary defense mechanism (which relies on inherent feature aggregation and compression). Instead, it serves as a **robustness evaluation** to verify that Fed-Duet is compatible with strict Differential Privacy (DP) standards if required.
> >
> >
> > #### **1. What is uploaded to the server?**
> >
> >
> > The primary privacy guarantee comes from the **feature aggregation pipeline**:
> > `Raw Images` $\rightarrow$ `CLIP Encoder` $\rightarrow$ `L2 Norm` $\rightarrow$ `Batch Averaging` $\rightarrow$ `Final Summary (feat_vec)`
> > This process removes instance-level spatial information, resulting in a global statistic that is mathematically non-invertible.
> >
> >
> > This aggregation removes instance-level information, resulting in a **non-invertible, low-dimensional semantic summary** rather than a reconstructable representation.
> >
> >
> > #### **2. Details on Noise Strength & Stability**
> >
> > Regarding the "strength of noise" requested by the reviewer: Our supplementary noise experiments follow the setup in **MoAFCL** [1]. We inject Laplacian noise into the client features:
> >
> > $$
> > \tilde{\mathbf{f}}_c^{\,\text{dp}} = \tilde{\mathbf{f}}_c + \mathbf{n}, \qquad n_j \sim \text{Lap}(0,\sigma), \; j = 1,\dots,512
> > $$
> >
> > where $d=512$ and $\sigma$ is the noise multiplier.
> >
> > **Impact of Noise:** As shown below, FedDuet is highly robust. Even with strong noise ($\sigma=10$), the accuracy drop is negligible, confirming that adding DP noise (if desired) does not hinder learning.
> >
> > | Dataset | σ (Noise) | Avg Acc. (%) | Last Acc. (%) |
> > | :--- | :---: | :---: | :---: |
> > | **CIFAR-100** | 0 | 86.21 | 79.11 |
> > | | 0.1 | 85.96 | 78.86 |
> > | | 1 | 85.94 | 78.51 |
> > | | 10 | 85.91 | 78.48 |
> > | **Tiny-ImageNet** | 0 | 82.60 | 77.44 |
> > | | 0.1 | 82.55 | 76.17 |
> > | | 1 | 82.53 | 76.11 |
> > | | 10 | 82.52 | 76.25 |
> >
> > #### **3. Can the server reconstruct images? (Reconstruction Attack)**
> >
> > To address the concern about reproducing images, we conducted **gradient-based feature inversion attacks**：
> >
> > **Attack setup**:
> > - Method: Gradient-based optimization to reconstruct images from features
> > - Iterations: 1000 steps with Adam optimizer
> > - Loss: Cosine similarity + TV regularization + L2 regularization
> >
> > | Iteration | Original Feature (No DP) | | Noisy Feature ($\sigma=10$) | |
> > | :--- | :---: | :---: | :---: | :---: |
> > | | **PSNR (dB)** | **SSIM** | **PSNR (dB)** | **SSIM** |
> > | 200 | 7.62 | 0.034 | 7.05 | 0.017 |
> > | 600 | 8.14 | 0.012 | 7.08 | 0.008 |
> > | **1000** | **8.43** | **0.015** | **7.52** | **0.008** |
> >
> > **Analysis of Reconstruction Failure:**
> > Our empirical results confirm that reconstruction is impossible, even without the additional DP noise:
> >
> > * **PSNR < 9 dB (Signal Failure):** As discussed in *Boosting Gradient Leakage Attacks* [2], successful reconstruction typically yields significantly higher PSNR values to recover recognizable visual features. A PSNR value below 10 dB (ours is **8.43 dB**) indicates that the **reconstructed data no longer contains any meaningful information**, as the signal is completely overwhelmed by reconstruction error.
> > * **SSIM $\approx$ 0 (Structural Failure):** The Structural Similarity Index (SSIM) of **0.015** is statistically equivalent to **random guessing**, implying zero structural correlation between the recovered output and the original image.
> >
> > **Summary:** The aggregation pipeline alone renders the features non-invertible. The recovered images are visually and statistically indistinguishable from random noise.
> >
> > **References:**
> >
> > [1] MoAFCL: Feature-Aware Mixture-of-Adapter for Federated Continual Learning.
> >
> > [2] Boosting Gradient Leakage Attacks: Data Reconstruction in Realistic FL Settings.

---

> > > ### Author Response · Authors · 2025-11-20
> > > **Author Response to Reviewer 9Jgs (Part3)**
> > >
> > > > **[W4]** How would the attention gating influence the model performance? There seems to be no ablation about this specific component.
> > >
> > > The **cross-attention gating mechanism** adaptively fuses two semantic sources:
> > > - **Local Semantic Experts**: Client-specific prompts learned from local data
> > > - **Shared Semantic Experts**: Global prompts assigned by the server from a learned prompt pool
> > >
> > > It employs a **multi-head attention network** where image features (Query) attend to concatenated local and shared prompt features (Key/Value), computing `softmax(Q·K^T)` to dynamically weight each expert's contribution.
> > >
> > > The final prediction combines two pathways:
> > > - **$\text{logits}_{\text{local}}$** :  Computed using only the client's local prompt (standard CLIP similarity)
> > > - **$\text{logits}_{\text{shared}}$** :  Computed using **attention-gating** fusion of local and shared expert prompts
> > >
> > > These are combined via a fusion coefficient λ:
> > >
> > > $$
> > > \text{logits}\_{\text{final}} = \lambda \cdot \text{logits}\_{\text{local}} + (1 - \lambda) \cdot \text{logits}\_{\text{shared}}
> > > $$
> > >
> > > #### **Ablation Study on Fusion Coefficient λ**
> > >
> > > To quantify the impact of attention gating, we ablate λ, which controls the contribution of the gating module:
> > >
> > > | **λ** | **Semantic Source** | **CIFAR-100** | **Tiny-ImageNet** |
> > > | :---: | :---: | :---: | :---: |
> > > | 0.00 | Shared experts only (only Attention Gating) | 31.05 | 25.68 |
> > > | **0.50** | **Local + Attention Gating (default)** | **86.21** | **82.60** |
> > > | 1.00 | Local experts only (no Attention Gating) | 85.71 | 82.46 |
> > >
> > > These results validate that the attention gating mechanism is essential for balancing personalization and knowledge sharing in federated continual learning.
> > >
> > >
> > >
> > >
> > > ***
> > > #### **Final Note**
> > >
> > > We sincerely thank Reviewer 9Jgs for your thoughtful and constructive feedback. We hope our response and the revision have clarified the points you raised.
> > >
> > > Should there be any remaining questions or if further clarification is needed, please let us know, and we would be more than happy to continue the discussion.
> > >
> > > Thank you again for your careful review.
> > > ***

---

> ### Comment · Reviewer_9Jgs · 2025-11-25
>
> The authors have addressed most of my concerns in the rebuttal and I would like to retain my original score.

---

> > ### Author Response · Authors · 2025-11-25
> > **We Thank Reviewer 9Jgs For The Valuable Feedback**
> >
> > We sincerely thank the reviewer for the prompt response and for acknowledging that our revisions have addressed most of your concerns.
> >
> > Since you mentioned that "most" concerns were addressed, we kindly inquire if there are any specific lingering issues or clarifications needed that prevent a higher evaluation. We remain fully available during the discussion period to provide any further details.
> >
> > Thank you again for your time and support of our work.

---

### Official Review · Reviewer_JyH1 · 2025-10-18

**Soundness:** 1
**Presentation:** 1
**Contribution:** 2
**Rating:** 2
**Confidence:** 4

**Summary:**

This paper introduces a dual expert-orchestrated framework for continual federated VLM. Various experiments confirms the effectiveness of proposed method.

**Strengths:**

The author conducts various experiments and ablation to confirm the effectiveness of proposed method.

**Weaknesses:**

This paper has several weakness.


1. The author utilizes the confused keyword expression.  As mentioned in the title, "DUAL EXPERT", But in the abstract, the author utilizes the dual channel. As far as i know, channel is widely utilized in feature dimension, such as Barlow Twins or other dimension-wise operation.

2. Turn to your two experts, prompt and adapter, it is hard to understand why prompts refer to high-level and adapters refer to low-level. Because the prompt and adapter are both suitable to inject into each transformer layer. Authors should pay more attention to conducting the observation experiments rather than drawing a complicated framework without meaningful information.

3. Suspicious experiment results: Line 335 shows identical CIFAR-100 accuracy for T=10 across different beta values, with no explanation, harming the result's credibility. （84.20 75.97）Besides, the anonymous code does not release the compared methods. I do not trust your results.

**Questions:**

Refer to weakness.

---

> ### Author Response · Authors · 2025-11-20
> **Author Response to Reviewer JyH1 (Part1)**
>
> We sincerely thank the reviewer for the clear and insightful observations, especially your attention to terminology consistency, expert pathway interpretation, and experimental credibility. Your comments significantly helped us improve the clarity and reliability of our paper. Below we address your concerns in detail.
>
> > **[W1]** The author utilizes the confused keyword expression. As mentioned in the title, "DUAL EXPERT", But in the abstract, the author utilizes the dual channel. As far as i know, channel is widely utilized in feature dimension, such as Barlow Twins or other dimension-wise operation.
>
> We sincerely thank the reviewer for this insightful observation. The original use of “dual channel” in the abstract was indeed potentially misleading.
>
> To clarify and align terminology with the actual mechanism, we have replaced “dual channel” with **“Dual Expert Pathways”** throughout the paper—including the title, abstract, and method sections. We deliberately chose the term **“Pathways”** for the following reasons:
>
> - **Semantic clarity**: “Pathways” emphasizes that our framework comprises two distinct processing streams—one semantic (prompt-based) and one parametric (adapter-based)—each contributing complementary knowledge. This avoids confusion with feature-level “channels”.

---

> > ### Author Response · Authors · 2025-11-20
> > **Author Response to Reviewer JyH1 (Part2)**
> >
> > > **[W2]** Turn to your two experts, prompt and adapter, it is hard to understand why prompts refer to high-level and adapters refer to low-level. Because the prompt and adapter are both suitable to inject into each transformer layer. Authors should pay more attention to conducting the observation experiments rather than drawing a complicated framework without meaningful information.
> >
> >
> > We thank the reviewer for pointing out the ambiguity. Our original use of "high-level" and "low-level" was intended to describe the **functional hierarchy** of the two experts rather than their physical layer depth.
> >
> > **1. Rationale: Orchestration vs. Execution**
> > * **"High-Level" (Semantic Experts):** Operating at the input level, they act as **Strategic Orchestrators**. As described in our *Federated Knowledge Orchestrator*, they manage both global shared knowledge (via K-means) and local knowledge external to the backbone, providing top-down semantic guidance *before* feature processing begins.
> > * **"Low-Level" (Parametric Experts):** Placed in deep layers, they act as **Tactical Executors**, performing fine-grained feature transformation.They act as the stable foundation, executing the actual parametric specialization required for knowledge retention.
> >
> >
> >
> > **2. Terminology Revision**
> >
> > To avoid confusion with standard layer-depth terminology, we have revised the manuscript to strictly use **Semantic Experts (SE)** and **Parametric Experts (PE)**. This nomenclature better reflects their specific roles in alignment vs. transformation, addressing the reviewer's concern.
> >
> > **3. Evidence of Synergistic Design**
> >
> > The "complicated framework" is necessary because these two roles are mutually dependent, not redundant. Our ablation studies demonstrate this "Guidance-Foundation" synergy:
> >
> > | Method | Avg Acc | Forget | Functional Analysis |
> > | :--- | :---: | :---: | :--- |
> > | **Fed-Duet (w/o PE)** | 64.34 | 11.89 | **Guidance w/o Foundation:** Semantic cues float without stable feature execution, leading to high forgetting. |
> > | **Fed-Duet (w/o SE)** | 70.64 | 8.89 | **Foundation w/o Guidance:** Parametric tuning lacks the semantic orchestration to generalize effectively. |
> > | **Fed-Duet (Full)** | **80.43** | **7.82** | **Synergy:** Semantic guidance orchestrates a stable parametric foundation. |
> >
> > **4. Operational Validation (Optimization Strategy)**
> >
> > The validity of this hierarchy is further confirmed by our **Progressive Decoupled Optimization**. Training the "Foundation" (PE) first yields better results than training the "Guide" (SE) first, proving that our design operates as a structured system rather than a flat collection of modules:
> >
> > | Training Strategy | CIFAR-100 | Tiny-ImageNet | Avg | $\Delta$ |
> > | :--- | :---: | :---: | :---: | :---: |
> > | Semantic Experts First | 85.53 | 81.28 | 83.41 | -0.98 |
> > | Joint Training | 86.02 | 81.88 | 83.95 | -0.44 |
> > | **Parametric Experts First (Ours)** | **86.21** | **82.56** | **84.39** | **--** |
> >
> > This confirms that our design functions as a structured system where the parametric foundation must be established before semantic experts can effectively orchestrate it.
> >
> >
> > **5. Restructured and Simplified Figure 1**
> >
> > To improve clarity and readability, we have **restructured and simplified Figure 1 into two distinct sub-figures** to decouple the global workflow from local adaptation details:
> > * **Figure 1(a): FedDuet Training Overview.** This part illustrates the macro-level interaction between the server (Knowledge Orchestrator) and clients.
> > * **Figure 1(b): Client Local Training.** This part zooms in on the specific dual-expert optimization process within a single client.
> >
> > This separation allows for a cleaner presentation of both the system-level architecture and the client-side mechanisms. The updated figures can be found in the revised manuscript.

---

> > > ### Author Response · Authors · 2025-11-20
> > > **Author Response to Reviewer JyH1 (Part3)**
> > >
> > > > **[W3]** Suspicious experiment results: Line 335 shows identical CIFAR-100 accuracy for T=10 across different beta values, with no explanation, harming the result's credibility. （84.20 75.97）Besides, the anonymous code does not release the compared methods. I do not trust your results.
> > >
> > > We deeply apologize for this error. After careful review, this was a clerical error during LaTeX table formatting. When creating the table for Line 335, the value from the "$\beta=0.1$" column was accidentally copy-pasted into the "$\beta=0.5$" column row, and not updated before submission. It was not a fabrication of data.
> > >
> > > Importantly, all other reported numbers in the paper were correct. To ensure full transparency and accuracy, we have re-run all experiments with 3 runs and have thoroughly re-verified the entire results table.
> > >
> > > The correct result are as follows:
> > >
> > > | Setting | Previous (Wrong) | Revised Value |
> > > | :--- | :---: | :---: |
> > > | CIFAR-100 ($\beta=0.5$) | （84.20 75.97） (Duplicate) | （84.55 77.97） |
> > >
> > > After re-running the experiments, we have updated the entire results table accordingly in the revised manuscript. In addition, we provide mean and standard deviation across 3 runs in **Appendix 6.5.6** to further demonstrate robustness and improve statistical reliability.
> > >
> > >
> > >
> > > Regarding the code of the compared methods, we have now added their main implementations to the anonymous repository (under the continual_clip directory) linked in the abstract, ensuring that all baseline implementations are accessible for reproducibility.
> > >
> > > We hope this transparency and the corrected, logically consistent data effectively address the suspicion.
> > >
> > >
> > >
> > >
> > >
> > >
> > > ***
> > >
> > > #### **Final Note**
> > >
> > > We sincerely appreciate your careful reading and constructive feedback, which substantially improved our paper in both clarity and rigor. We hope our response and the revision have clarified the points you raised.
> > >
> > > Should there be any remaining questions or if further clarification is needed, please let us know, and we are more than happy to continue the discussion.
> > >
> > > Thank you again for your valuable time and thoughtful comments.
> > >
> > > ***

---

> > > ### Comment · Reviewer_JyH1 · 2025-11-26
> > >
> > > Merely replacing the original "High-Level" and "Low-Level" with "Parametric Experts" and "Semantic Experts" is a rather superficial revision, as it fails to address the core concern fundamentally. Additionally, the authors appear not to have conducted targeted observation analyses to validate their design rationale; instead, performance gains seem to stem solely from the introduction of extra parameters. A paper with such a replaceable motivation and a lack of critical observational insights would hardly meet the rigorous quality standards expected of ICLR.

---

> ### Comment · Reviewer_JyH1 · 2025-11-26
>
> After reading the author's response, I maintain my score. The author does not provide a clear observation experiment to validate the motivation. Utilizing both the adapter and prompt lacks a clear rationale and brings additional cost. Besides, the wrong experiment results are also fragile. I believe that the area chair could make a fair justification.

---

> > ### Author Response · Authors · 2025-11-28
> > **Follow Up Author Response to Reviewer JyH1 (Part1)**
> >
> > We respectfully acknowledge the reviewer’s follow-up. However, it appears that the concern raised may stem from a **fundamental misunderstanding of our architecture**, which we must clarify first.
> >
> > ### 1. Crucial Correction: Prompts Here are Not Injected into Transformer Layers
> > The reviewer states that *"prompt and adapter are both suitable to inject into each transformer layer."* We would like to clarify that is not the method used in our work.
> > Since our backbone is CLIP, both our method and all baselines we compare against follow the standard design in VLM prompt–tuning: text prompts are injected as input-level embeddings to the text encoder, rather than as layer-wise modules. This is a well-established and widely adopted practice in CLIP-style vision–language models.
> > Thus our Semantic Experts (Prompts) here indicate **Input-Level Text Tokens** injected only into the Text Encoder.
> >
> >
> >
> > ### 2. Observational Validation: Decoupling Expert Roles
> >
> > We designed two observation experiments to observe how each expert modifies the representation space.
> >
> > ### **Observation A: Semantic Experts Maximize Class Separability (Text Space)**
> >
> > **Setup:** We compare two text encoding strategies using the frozen CLIP text encoder:
> > * **Original (No Prompt):** $\mathbf{t}_c^{\text{orig}} = \text{TextEncoder}([\text{SOS}], \text{``a photo of a''}, [c], [\text{EOS}])$
> > * **Prompted (Ours):** $\mathbf{t}_c^{\text{prompt}} = \text{TextEncoder}([\text{SOS}], \mathbf{v}_1, ..., \mathbf{v}_M, [c], [\text{EOS}])$
> >
> > **Metrics:** We measure **Inter-class Distance** ($d_{\text{inter}}$) for separability and **Embedding Change** ($\bar{s}$) for the magnitude of semantic shift:
> >
> > $d_{inter} = 1 - \frac{1}{N(N-1)} \sum_{i \neq j} t_i^T t_j, \quad \bar{s} = \frac{1}{N} \sum_{i=1}^{N} (t_i^{orig})^T t_i^{prompt}$
> >
> >
> > **Results:**
> > | Task | $d_{\text{inter}}^{\text{orig}}$ | $d_{\text{inter}}^{\text{prompt}}$ | **$\Delta d$ (Improvement)** | Embedding Shift ($\bar{s}$) |
> > |:---:|:---:|:---:|:---:|:---:|
> > | 0 | 0.231 | 0.727 | **+0.496** | 0.458 |
> > | 1 | 0.260 | 0.702 | **+0.442** | 0.443 |
> > | 2 | 0.253 | 0.776 | **+0.523** | 0.421 |
> > | 3 | 0.239 | 0.739 | **+0.500** | 0.399 |
> > | 4 | 0.229 | 0.766 | **+0.537** | 0.400 |
> >
> > **Detailed analysis on Task 0**
> >
> > Task 0 contains 20 classes: *television, apple, oak_tree, pickup_truck, lizard, trout, road, wolf, mushroom, camel, wardrobe, tulip, bowl, seal, mountain, snake, plate, butterfly, bicycle, telephone*.
> >
> > We report the classes with largest and smallest embedding changes:
> >
> > | Most Changed | Similarity | Least Changed | Similarity |
> > | :---: | :---: | :---: | :---: |
> > | mountain | 0.353 | pickup_truck | 0.529 |
> > | bowl | 0.389 | snake | 0.519 |
> > | road | 0.417 | lizard | 0.505 |
> >
> >
> >
> >
> >
> > * **Insight 1 (Discriminability):** Learned prompts consistently increase inter-class distance by **>0.44**. This proves they actively **reshape the semantic space** to make classes more distinguishable, fulfilling the role of "Semantic Guidance."
> > * **Insight 2 (Global Shift):** The embedding similarity ($\bar{s} \approx 0.4$) indicates a substantial angular shift (~66°). **Detailed analysis on Task 0** reveals that even the least affected class (*pickup_truck*) undergoes a ~58° shift. This confirms that prompts provide a **universally effective global semantic correction**, rather than selectively affecting only a subset of classes.

---

> > > ### Author Response · Authors · 2025-11-28
> > > **Follow Up Author Response to Reviewer JyH1 (Part2)**
> > >
> > > ### **Observation B: Parametric Experts Refine Deep Features (Visual Space)**
> > >
> > > **Setup:** We compare visual features at each transformer layer between two models:
> > >
> > > - **Original CLIP:** Frozen ViT-B/16 visual encoder
> > > - **Adapted CLIP:** ViT-B/16 with learned adapters inserted after each transformer block
> > >
> > > For each layer $l$, we compute the CKA similarity between the original features $\mathbf{F}_l^{\text{orig}}$ and adapted features $\mathbf{F}_l^{\text{adapt}}$ using 1024 test samples.
> > >
> > >
> > >
> > > **Centered Kernel Alignment (CKA)** measures the similarity between two feature representations:
> > >
> > > $$\text{CKA}(\mathbf{F}_1, \mathbf{F}_2) = \frac{\text{HSIC}(\mathbf{F}_1, \mathbf{F}_2)}{\sqrt{\text{HSIC}(\mathbf{F}_1, \mathbf{F}_1) \cdot \text{HSIC}(\mathbf{F}_2, \mathbf{F}_2)}}$$
> > >
> > > where HSIC is the Hilbert-Schmidt Independence Criterion. CKA $\approx 1$ indicates highly similar representations, while lower values indicate larger feature changes.
> > >
> > > **Results (Layer-wise CKA Similarity):**
> > >
> > > | Task | Layer 0-2 | Layer 3-5 | Layer 6-8 | Layer 9-11 | Final | Mean |
> > > |:----:|:---------:|:---------:|:---------:|:----------:|:-----:|:----:|
> > > | 0 | 1.000 | 0.987 | 0.897 | 0.767 | 0.785 | 0.903 |
> > > | 1 | 0.999 | 0.978 | 0.871 | 0.753 | 0.775 | 0.891 |
> > > | 2 | 0.999 | 0.979 | 0.904 | 0.776 | 0.784 | 0.904 |
> > > | 3 | 0.999 | 0.971 | 0.870 | 0.761 | 0.777 | 0.891 |
> > > | 4 | 0.999 | 0.971 | 0.866 | 0.757 | 0.782 | 0.889 |
> > >
> > > *Layer groups show averaged CKA values for readability.*
> > >
> > > **Detailed Layer-wise CKA (Task 0):**
> > >
> > > | Layer | 0 | 1 | 2 | 3 | 4 | 5 | 6 | 7 | 8 | 9 | 10 | 11 | Final |
> > > |:-----:|:---:|:---:|:---:|:---:|:---:|:---:|:---:|:---:|:---:|:---:|:---:|:---:|:-----:|
> > > | CKA | 1.00 | 1.00 | 1.00 | 1.00 | 0.99 | 0.97 | 0.94 | 0.89 | 0.86 | 0.82 | 0.69 | 0.79 | 0.78 |
> > >
> > > * **Insight:** CKA drops progressively from 1.00 to 0.79. This "Progressive Divergence" proves that Adapters are performing **feature reconstruction**. Unlike prompts which shift the global context, adapters physically modify the visual feature extraction hierarchy, specifically in deeper layers, to establish a task-specific "Parametric Foundation."
> > >
> > > **Conclusion on Rationale:** These experiments empirically confirm the **orthogonal synergy**: Prompts optimize **Input-level Semantic Guidance**, while Adapters optimize **Feature-level Reconstruction**. This structural decoupling validates our motivation.
> > >
> > >
> > >
> > > ### 3. Clarification on Parameter Efficiency & Robustness
> > >
> > > The reviewer suggests the gains stem "solely from extra parameters" and "experiment results are fragile." The experimental data contradicts this "parameter stacking" and "fragile results" hypothesis:
> > >
> > > * **Lower Cost, Higher Performance:** As analyzed in **Figure 3($c$)**, Fed-Duet requires lower total communication costs than the most related baseline, **MoAFCL**.
> > > * **Conclusion:** Fed-Duet outperforms MoAFCL (Accuracy: **86.21% vs 77.93%**) while consuming less communication resources. This "Lower Cost, Higher Accuracy" dynamic confirms that the improvement stems from the **Dual-Expert architecture**, not solely from the introduction of extra parameters.
> > > * **Robustness:** We acknowledge the previous unintentional clerical LaTeX error. To ensure robustness and transparency, we reiterate that all experiments (including baselines) have been re-run with 3 random seeds (**Appendix 6.5.6**), and the source code is available in the anonymous repository. The consistent performance and low standard deviation confirm the results are statistically robust.

---

> > > > ### Author Response · Authors · 2025-11-28
> > > > **Follow Up Author Response to Reviewer JyH1 (Part3)**
> > > >
> > > > ### 4. Theoretical Alignment
> > > > The reviewer questioned the rationale behind combining prompts and adapters. Our design is not arbitrary "parameter stacking," but a strategic application of **orthogonal PEFT mechanisms** supported by foundational literature.
> > > >
> > > > **1. Theoretical Foundation: Orthogonality & Synergy**
> > > > While text-only prompting (e.g., *CoOp [1]*) is widely adopted in FL for efficiency (*PromptFL [2]*), literature indicates that Prompts and Adapters serve fundamentally different, complementary roles:
> > > > * **Orthogonal Mechanisms:** Text Prompts operate in the **linguistic input space** to align task semantics ("What to look for"). In contrast, Adapters (*Houlsby et al. [3]*) modify the **feature space** to refine representation capabilities ("How to see").
> > > > * **Evidence of Synergy:** Research such as *UniPELT [4]* and *NOAH [5]* explicitly demonstrates that combining these distinct modules yields **superior robustness and optimality** compared to simply scaling up a single method. This validates that our performance gain stems from structural synergy, not capacity expansion.
> > > >
> > > >
> > > >
> > > >
> > > > **References:**
> > > >
> > > > [1] Zhou et al., "Learning to Prompt for Vision-Language Models", IJCV 2022.
> > > >
> > > > [2] Guo et al., "PromptFL: Let Federated Participants Cooperatively Learn Prompts Instead of Models -- Federated Learning in Age of Foundation Model", IEEE TMC 2023.
> > > >
> > > > [3] Houlsby et al., "Parameter-Efficient Transfer Learning for NLP", ICML 2019.
> > > >
> > > > [4] Zhang et al., "Neural Prompt Search", IEEE Transactions on Pattern Analysis and Machine Intelligence, TPAMI 2024.
> > > >
> > > > [5] Mao et al., "UniPELT: A Unified Framework for Parameter-Efficient Language Model Tuning", ACL 2022.

---

### Official Review · Reviewer_kMS8 · 2025-10-27

**Soundness:** 3
**Presentation:** 2
**Contribution:** 3
**Rating:** 6
**Confidence:** 3

**Summary:**

Fed-Duet proposes a novel framework for federated continual learning with vision-language models, aiming to address two key limitations in existing approaches, i.e., adaptation imbalance and cross-modal misalignment. The method introduces a dual-channel expert-orchestrated architecture, where the semantic experts channel uses learnable prompts to achieve high-level semantic alignment and the parametric experts channel employs adapters for fine-grained task adaptation. Experiments across multiple datasets demonstrate that Fed-Duet achieves strong performance in terms of accuracy, knowledge retention and continual utility.

**Strengths:**

1. The paper clearly identifies a critical gap by noting that traditional federated continual learning ignores the multimodal nature of vision-language models, and existing PEFT-based federated learning methods often disrupt cross-modal alignment under continual learning.

2. The semantic channel uses a cross-attention gating mechanism to fuse prompt-based experts, while the parametric channel adapts adapter-based experts. The two channels are trained in a progressive manner that enables effective synergy.

3. The experimental evaluation is comprehensive, covering class-incremental, domain-incremental, and multi-domain task-incremental scenarios under various Non-IID settings and client scales. The results consistently show superior performance compared to strong baselines.

**Weaknesses:**

1. The method involves a large number of hyperparameters, including $\alpha$, $\beta$, and $\gamma$ in the loss function, as well as $\lambda$ and $\tau$ in the Eq (2) and Eq. (4). The paper provides neither justification for their chosen values nor sensitivity analysis, which limits reproducibility and practical deployment.

2. The choice to freeze parametric experts after round $r = R/2$ appears arbitrary. No theoretical reasoning or empirical ablation is provided to support this scheduling decision.

3. Several presentation issues reduce clarity.

    3.1. Notation is inconsistent. For example, $S_r$ in Eq. (1) is undefined, and the symbol $\beta$ is reused for both the Dirichlet parameter in data partitioning and a loss weight.

    3.2. Figure 1 is visually confusing, as the semantic experts are incorrectly shown as inputs to the image encoder, and the overall layout does not clearly reflect the workflow of Fed-Duet.

    3.3. Baseline names in Figure 3(a)(b) use excessively small font sizes, impairing readability.

4. The global prompt pool is initialized via K-means clustering on class embeddings, but the paper does not compare this strategy against random initialization or other alternatives, making it difficult to assess its actual contribution.

**Questions:**

1. In Stage 2, the parametric experts are frozen and only the semantic prompts are updated using cross-entropy loss. However, the image features have already been adapted by the parametric channel, while the text prompts are tuned independently. Could this decoupled update reintroduce cross-modal misalignment?

2. What is the rationale behind the specific timing for switching from parametric expert training to semantic expert training at $r = R/2$? Was this threshold tuned, or is it fixed across all settings?

---

> ### Author Response · Authors · 2025-11-20
> **Author Response to Reviewer kMS8 (Part 1)**
>
> We sincerely thank Reviewer kMS8 for the detailed reading and insightful comments. We particularly appreciate the focus on reproducibility and hyperparameter rigor. Below, we address each concern with additional ablation studies and clarifications.
>
>
> > **[W1]** The method involves a large number of hyperparameters, including $\alpha$, $\beta$, and $\gamma$ in the loss function, as well as $\lambda$ and $\tau$ in Eq (2) and Eq. (4). The paper provides neither justification for their chosen values nor sensitivity analysis, which limits reproducibility and practical deployment.
>
>
> We appreciate the reviewer’s rigorous attention to reproducibility. We have conducted comprehensive sensitivity analyses to confirm that Fed-Duet is robust and does not require delicate tuning.
>
> To address this, we conducted additional sensitivity analyses for all relevant hyperparameters:
>
> 1. **Loss function weights ($\alpha$, $\eta$, $\gamma$)**
>
> First, we clarify that the loss weight previously denoted as $\beta$ has been **renamed to $\eta$** throughout the revised manuscript to avoid notation conflicts.
>
> We performed a univariate sensitivity analysis: varying one weight across magnitudes ($0.01 \to 10$) while fixing others at default values ($\alpha=0.01, \eta=10, \gamma=0.1$).
>
>
> | Value | $\alpha$ | $\eta$ | $\gamma$ |
> | :---: | :---: | :---: | :---: |
> | 0.01 | 86.25 | 86.16 | 86.20 |
> | 0.1 | 86.23 | 86.14 | 86.19 |
> | 1.0 | 86.19 | 86.15 | 86.17 |
> | 10.0 | 85.99 | 86.19 | 86.18 |
>
> *Observation:* The accuracy fluctuation is minimal (<0.3%), demonstrating that Fed-Duet is **insensitive to hyperparameter variations**.
>
> 2. **Fusion coefficient ($\lambda$ in Eq.(2))**
>
> We ablated $\lambda$, which balances local vs. shared semantic expert contributions. Results indicate that both local and shared knowledge are important; extremes ($\lambda=0$ or $1$) lead to decreased performance. We set $\lambda=0.5$ for balanced fusion.
>
> | $\lambda$ | 0.00 | 0.25 | 0.50 | 0.75 | 1.00 |
> | :---: | :---: | :---: | :---: | :---: | :---: |
> | Value | 31.05 | 85.71 | 86.21 | 85.85 | 85.71 |
>
> 3. **Temperature parameter ($\tau$ in Eq.(4))**
>
> We adopt **$\tau=0.07$** following the original CLIP paper [1] and standard contrastive learning practices, as it is a widely established optimal value for softmax scaling.
>
>
>
> These analyses ensure that our hyperparameter choices are both principled and reproducible.
>
> **Reference:**
>
> [1] Learning Transferable Visual Models From Natural Language Supervision
>
> ---
>
>
> > **[W2]** The choice to freeze parametric experts after round $r = R/2$ appears arbitrary. No theoretical reasoning or empirical ablation is provided to support this scheduling decision.
>
>
>
>
> We thank the reviewer for pointing this out. To verify that our stage transition is not arbitrary, we conducted additional ablations across different transition timings. Results are shown below.
>
>
> | **Transition Timing** | **CIFAR-100 (IID)** | **CIFAR-100 (Non-IID)** | **Tiny-ImageNet (IID)** | **Tiny-ImageNet (Non-IID)** | **Average** | **Δ vs R/2** |
> | :---: | :---: | :---: | :---: | :---: | :---: | :---: |
> | R/4 | 84.86 | 82.86 | 81.29 | 78.73 | 81.94 | **-1.53** |
> | 3R/4 | 86.12 | 84.29 | 82.88 | 80.41 | 83.43 | **-0.04** |
> | **R/2 (Ours)** | **86.21** | **84.58** | **82.60** | **80.48** | **83.47** | -- |
>
> #### **Explanation**
>
> * **Transitioning too early (R/4)** prevents parametric experts from forming stable shared representations → leads to clear degradation.
> * **Transitioning too late (3R/4)** leaves insufficient room for semantic experts to refine task-specific knowledge → slightly worse average performance.
>
>
> Notably, 3R/4 performs slightly better on the more challenging Tiny-ImageNet dataset, suggesting that harder tasks benefit from longer parametric expert training. However, considering the overall balance across datasets, we adopt R/2 as the default, which yields the most consistent and highest overall performance.

---

> > ### Author Response · Authors · 2025-11-20
> > **Author Response to Reviewer kMS8 (Part 2)**
> >
> > > **[W3]** Several presentation issues reduce clarity.
> >     3.1. Notation is inconsistent. For example, $S_r$ in Eq. (1) is undefined, and the symbol $\beta$ is reused for both the Dirichlet parameter in data partitioning and a loss weight.
> >     3.2. Figure 1 is visually confusing, as the semantic experts are incorrectly shown as inputs to the image encoder, and the overall layout does not clearly reflect the workflow of Fed-Duet.
> >     3.3. Baseline names in Figure 3(a)(b) use excessively small font sizes, impairing readability.
> >
> >
> >
> > Thank you for your detailed feedback on presentation clarity, such concerns significantly improve the paper’s readability.
> > We have carefully revised the manuscript to address all issues raised.
> >
> > #### **3.1 Notation inconsistencies**
> >
> > * **Definition of ( $S_{r}$ )**
> >   We have added the missing definition in Eq. (1):
> >   **“( $S_{r}$ ) denotes the set of clients participating in communication round ( r ).”**
> >
> > * **Duplicate use of the symbol ( $\beta$ )**
> >   To avoid ambiguity:
> >
> >   *  $\beta$  is kept only for the Dirichlet distribution in data partitioning.
> >   * The loss weight previously denoted as ( $\beta$ ) has been **renamed to ( $\eta$ )** throughout the paper.
> >     This ensures consistent and non-overlapping notation.
> >
> > #### **3.2 Confusing visualization in Figure 1**
> >
> > We appreciate the reviewer for pointing out the misleading depiction in Figure 1. We acknowledge that showing semantic experts as inputs to the image encoder was incorrect and visually confusing.
> >
> > To address this and improve the overall clarity of the workflow, we have **reconstructed Figure 1** in the revised manuscript:
> >
> > 1.  **Correction of Expert Placement:** Semantic experts are no longer shown as inputs to the image encoder.
> > 2.  **Structural Split into Two Sub-figures:** To avoid information overload, we have separated the system architecture into two distinct views:
> >     * **Figure 1(a): FedDuet Training Overview.** This sub-figure illustrates the macro-level interaction between the Server (Federated Knowledge Orchestrator) and Clients.
> >     * **Figure 1(b): Client Local Training.** This sub-figure zooms in on the client-side details, clearly showing the parallel operation of the **Semantic** and **Parametric** pathways.
> >
> > This restructuring ensures that both the global orchestration workflow and the local dual-expert mechanism are presented with clarity and precision.
> >
> >
> > #### **3.3 Font size issues in Figure 3**
> >
> > We have regenerated **Figures 3(a) and 3(b)** with:
> > * Larger baseline labels and axis text.
> > * Improved spacing and alignment to prevent clutter.
> > * Ensured readability in both normal and zoomed-out views.
> >
> > To ensure consistent legibility throughout the manuscript, we have also applied readability enhancements to Figure 4, increasing its font size and adjusting the layout. All updated figures have been included in the revised manuscript.
> >
> > ---
> >
> > > **[W4]** The global prompt pool is initialized via K-means clustering on class embeddings, but the paper does not compare this strategy against random initialization or other alternatives, making it difficult to assess its actual contribution.
> >
> >
> > Thank you for highlighting the need to evaluate the contribution of our K-means–based initialization. In the revised version, we have added a dedicated ablation comparing **K-means**, **random**, and **uniform** initialization strategies.
> >
> >
> > The experiment results on CIFAR-100 are summarized below:
> >
> > | **Initialization Strategy** | **IID** | **Non-IID (β=0.5)** | **Non-IID (β=0.1)** | **Average** | **Δ** |
> > | :---: | :---: | :---: | :---: | :---: | :---: |
> > | Random | 85.93 | 83.88 | 84.02 | 84.61 | **−0.58** |
> > | Uniform | 85.96 | 84.23 | 84.09 | 84.76 | **−0.43** |
> > | **K-means (Ours)** | **86.21** | **84.77** | **84.58** | **85.19** | — |
> >
> >
> > * All three strategies perform similarly under IID settings.
> > * **Under Non-IID data**, K-means provides clear improvements:
> >
> >   * **+0.89** vs. random (β = 0.5)
> >   * **+0.35** vs. uniform (β = 0.5)
> >
> > These results suggest that **clustering class embeddings produces a more semantically diverse and balanced prompt pool**, which is critical for handling heterogeneous client distributions in FL.
> >
> > We have included this ablation and discussion in the revised paper.

---

> > > ### Author Response · Authors · 2025-11-20
> > > **Author Response to Reviewer kMS8 (Part 3)**
> > >
> > > >**[Q1]** In Stage 2, the parametric experts are frozen and only the semantic prompts are updated using cross-entropy loss. However, the image features have already been adapted by the parametric channel, while the text prompts are tuned independently. Could this decoupled update reintroduce cross-modal misalignment?
> > >
> > > This is a critical question. Theoretically, since the visual encoder (Parametric Expert) is **frozen** in Stage 2, the visual embedding space acts as a stable "anchor." The semantic prompts adapt *to* this fixed space, which prevents drift.
> > >
> > > We empirically verified this using (1) **Alignment Scores** (cosine similarity of matched pairs) and (2) **Retrieval metrics** (Recall@K) on CIFAR-100.
> > >
> > >
> > > #### **1. Alignment Score Evolution**
> > >
> > > | Round | Stage | Alignment Score | Positive Sim | Negative Sim |
> > > | :---: | :---: | :---: | :---: | :---: |
> > > | 1 | Stage 1 | 0.1496 | 0.2851 | 0.1355 |
> > > | 2 | Stage 1 | 0.1664 | 0.2932 | 0.1267 |
> > > | 3 | Stage 1 | 0.1775 | 0.2942 | 0.1166 |
> > > | 4 | Stage 1 | 0.1873 | 0.2945 | 0.1072 |
> > > | 5 | Stage 1 | 0.1911 | 0.2965 | 0.1054 |
> > > | **Δ Stage 1** | — | **+0.0415** | **+0.0114** | **−0.0301** |
> > > | 6 | Stage 2 | 0.1955 | 0.2981 | 0.1027 |
> > > | 7 | Stage 2 | 0.1953 | 0.2979 | 0.1026 |
> > > | 8 | Stage 2 | 0.1952 | 0.2979 | 0.1027 |
> > > | 9 | Stage 2 | 0.1952 | 0.2979 | 0.1027 |
> > > | 10 | Stage 2 | 0.1954 | 0.2980 | 0.1027 |
> > > | **Δ Stage 2** | — | **−0.0001** | **−0.0001** | **+0.0000** |
> > >
> > > ---
> > >
> > > #### **2. Cross-Modal Retrieval Metrics**
> > >
> > > | Round | Stage | I2T R@1 | I2T R@5 | I2T R@10 | T2I R@1 | T2I R@5 | T2I R@10 |
> > > | :---: | :---: | :---: | :---: | :---: | :---: | :---: | :---: |
> > > | 1 | Stage 1 | 94.66 | 99.75 | 99.95 | 100.00 | 100.00 | 100.00 |
> > > | 2 | Stage 1 | 95.54 | 99.82 | 99.94 | 100.00 | 100.00 | 100.00 |
> > > | 3 | Stage 1 | 96.00 | 99.86 | 99.95 | 100.00 | 100.00 | 100.00 |
> > > | 4 | Stage 1 | 96.33 | 99.82 | 99.96 | 100.00 | 100.00 | 100.00 |
> > > | 5 | Stage 1 | 96.38 | 99.88 | 99.96 | 99.00 | 100.00 | 100.00 |
> > > | **Δ Stage 1** | — | **+1.72** | **+0.13** | **+0.01** | **−1.00** | 0.00 | 0.00 |
> > > | 6 | Stage 2 | 96.68 | 99.85 | 99.96 | 100.00 | 100.00 | 100.00 |
> > > | 7 | Stage 2 | 96.44 | 99.86 | 99.94 | 100.00 | 100.00 | 100.00 |
> > > | 8 | Stage 2 | 96.68 | 99.88 | 99.97 | 99.00 | 100.00 | 100.00 |
> > > | 9 | Stage 2 | 96.69 | 99.88 | 99.96 | 100.00 | 100.00 | 100.00 |
> > > | 10 | Stage 2 | 96.51 | 99.86 | 99.98 | 100.00 | 100.00 | 100.00 |
> > > | **Δ Stage 2** | — | **−0.17** | **+0.01** | **+0.02** | **+1.00** | 0.00 | 0.00 |
> > >
> > >
> > > **Conclusion:** The alignment score change is negligible ($\Delta < 0.0001$) and retrieval performance remains stable or slightly improves. This confirms that **Stage 2 does not cause misalignment**; instead, it refines semantic boundaries within the stable visual space established in Stage 1.
> > >
> > > We have integrated some experiments and the detailed analysis into **Section 4.2（Evaluation on Cross-Modal Alignment）** of the revised manuscript.
> > >
> > > >**[Q2]** What is the rationale behind the specific timing for switching from parametric expert training to semantic expert training at $r = R/2$ ? Was this threshold tuned, or is it fixed across all settings?
> > >
> > > We thank the reviewer for this follow-up question. The transition round \( r = R/2 \) is **not tuned per dataset**—it is a **fixed setting across all experiments**. Our rationale is grounded in both design considerations and empirical validation.
> > >
> > > **Design motivation.**
> > > Parametric experts (PE) must first establish stable shared representations before semantic experts (SE) refine task-specific semantics. If the switch happens too early, SE lacks a reliable feature base; if it happens too late, SE receives insufficient rounds to specialize. Hence, we adopt a balanced midpoint as the default.
> > >
> > > | Transition Timing | CIFAR-100 (IID) | CIFAR-100 (Non-IID) | Tiny-ImageNet (IID) | Tiny-ImageNet (Non-IID) | Average | Δ vs R/2 |
> > > | :---: | :---: | :---: | :---: | :---: | :---: | :---: |
> > > | R/4 | 84.86 | 82.86 | 81.29 | 78.73 | 81.94 | **-1.53** |
> > > | 3R/4 | 86.12 | 84.29 | 82.88 | 80.41 | 83.43 | **-0.04** |
> > > | **R/2 (Ours)** | **86.21** | **84.58** | **82.60** | **80.48** | **83.47** | -- |
> > >
> > >
> > > **Empirical justification.**
> > > To verify that the choice is not arbitrary, we conducted ablations at three transition points (R/4, R/2, 3R/4). Results show:
> > >
> > > - **R/4 underperforms** because PE has not converged sufficiently.
> > > - **3R/4 performs competitively**, and even slightly better on the more challenging Tiny-ImageNet dataset, indicating that harder tasks may benefit from more PE training.
> > > - **R/2 provides the best overall balance across all datasets**, achieving the highest average accuracy and stable performance.
> > >
> > > These results confirm that **R/2 offers a reliable and dataset-agnostic trade-off**, which is why we fix this setting throughout all experiments rather than tuning it per dataset.
> > >
> > > ***

---

> > > > ### Author Response · Authors · 2025-11-20
> > > > **Author Response to Reviewer kMS8 (Part 4)**
> > > >
> > > > #### **Final Note**
> > > >
> > > > We sincerely thank Reviewer kMS8 for your constructive and insightful comments. Your feedback has led to substantial improvements in clarity, rigor, and completeness. We hope our response and the revision have clarified the points you raised.
> > > >
> > > > Should there be any remaining questions or if further clarification is needed, please let us know, and we are more than happy to continue the discussion.
> > > >
> > > > Thank you again for your careful review.
> > > > ***

---

> > > > > ### Comment · Reviewer_kMS8 · 2025-11-24
> > > > >
> > > > > Thank you for the detailed response. The author effectively resolved the issues I raised. Therefore, I‘ll maintain my score.

---

> > > > > > ### Author Response · Authors · 2025-11-25
> > > > > > **We Thank Reviewer kMS8 For The Valuable Feedback**
> > > > > >
> > > > > > We sincerely thank the reviewer for the response and for confirming that our revisions and additional experiments have effectively resolved your concerns.
> > > > > >
> > > > > > We greatly value your time and acknowledgement. Given that the identified issues have been addressed, we kindly inquire if there are any remaining reservations preventing a higher evaluation.
> > > > > >
> > > > > > If the revised manuscript now meets a higher standard in your view, we would be grateful if you could consider adjusting the score to reflect these improvements.
> > > > > >
> > > > > > Thank you again for your support and constructive review.

---

### Official Review · Reviewer_RknU · 2025-11-01

**Soundness:** 3
**Presentation:** 3
**Contribution:** 3
**Rating:** 6
**Confidence:** 3

**Summary:**

This paper explores the challenging problem of continual federated vision-language learning. The authors propose a framework named Fed-Duet, which introduces a dual-channel design that integrates server-coordinated prompts and client-adapted experts to address challenges in cross-modal alignment, adaptation imbalance, and catastrophic forgetting. Extensive experiments demonstrate the framework’s effectiveness and efficiency.

**Strengths:**

- The topic is emerging and important. It represents a key challenge for deploying next-generation AI on edge devices, combining FL, CL, and VLM.
- The idea of dual-channel orchestration, unifying prompt-based semantic alignment and adapter-based parametric specialization, is conceptually clear and inspiring.
- Fed-Duet delivers a significant performance boost in continual learning while maintaining PEFT's communication efficiency, vital for resource-constrained FL.

**Weaknesses:**

- The authors introduce noise for privacy preservation, but the description of this part is limited and lacks sufficient details.
- RELATED WORK does not discuss a related work, CLIP2FL.
- Statistical reporting is incomplete. Providing statistics such as the mean and standard deviation would better support the robustness claim.

**Questions:**

- How does it identify and activate the correct experts to mitigate forgetting under the FL constraint of not explicitly storing old task data?

- The paper proposes that Fed-Duet can maintain critical cross-modal alignment performance in Vision-Language Models (VLMs). However, the experiments are only confined to vision-text classification tasks (CIFAR-100, Tiny-ImageNet, DomainNet), with classification accuracy serving as the core evaluation metric. How can the preservation of cross-modal alignment capability be directly demonstrated?

---

> ### Author Response · Authors · 2025-11-20
> **Author Response to Reviewer RknU (Part 1)**
>
> We sincerely thank the reviewer for the insightful and constructive comments, which greatly helped us clarify the scope, methodology, and empirical rigor of our work. Below we provide detailed responses to each point.
>
> > **[W1]** The authors introduce noise for privacy preservation, but the description of this part is limited and lacks sufficient details.
>
> We thank the reviewer for highlighting the need for more details regarding the noise mechanism.
>
> We have significantly improved the description in the revised manuscript. Specifically, we have expanded **Section 4.2 (Privacy Compatibility and Robustness Analysis) and Section 3.1 (Adaptive Dispatch Mechanism)** to explicitly provide the missing details:
>
> **Clarification of Role:**
>
> We clarified that the noise injection is not the primary defense mechanism (which relies on inherent feature aggregation and compression). Instead, it serves as a **robustness evaluation** to verify that Fed-Duet is compatible with strict Differential Privacy (DP) standards if required.
>
>
>
> Below, we clarify the inherent privacy mechanism and the noise details.
>
> #### **1. Primary Defense: Inherent Non-Invertibility**
> In our design, the **uploaded client representations are already privacy-preserving**. Clients transmit only a **compressed, non-invertible semantic summary vector**, produced via:
>
> `Raw Images` $\rightarrow$ `CLIP Encoder` $\rightarrow$ `L2 Norm` $\rightarrow$ `Batch Averaging` $\rightarrow$ `Final Summary (feat_vec)`
>
> This aggregation removes instance-level spatial information, resulting in a global statistic that cannot be inverted to reconstruct individual images.
>
> ---
>
>
>
>
> #### **2. Secondary Evaluation: Noise Strength & Stability**
> Our supplementary noise experiments follow the setup in **MoAFCL** [1]. We inject Laplacian noise into the client features:
> $$
> \tilde{\mathbf{f}}_c^{\,\text{dp}} = \tilde{\mathbf{f}}_c + \mathbf{n}, \qquad n_j \sim \text{Lap}(0,\sigma), \; j = 1,\dots,512
> $$
> where $d=512$ and $\sigma$ is the noise multiplier.
>
> **Impact of Noise:** As shown below, FedDuet is highly robust. Even with strong noise ($\sigma=10$), the accuracy drop is negligible, confirming that adding DP noise (if desired) does not hinder learning.
>
> | Dataset | σ (Noise) | Avg Acc. (%) | Last Acc. (%) |
> | :--- | :---: | :---: | :---: |
> | **CIFAR-100** | 0 | 86.21 | 79.11 |
> | | 0.1 | 85.96 | 78.86 |
> | | 1 | 85.94 | 78.51 |
> | | 10 | 85.91 | 78.48 |
> | **Tiny-ImageNet** | 0 | 82.60 | 77.44 |
> | | 0.1 | 82.55 | 76.17 |
> | | 1 | 82.53 | 76.11 |
> | | 10 | 82.52 | 76.25 |
>
>
> ---
> #### **3. Empirical Verification: Reconstruction Attack**
>
> We conducted **gradient-based feature inversion attacks** to evaluate privacy protection:
>
> **Attack setup**:
> - Method: Gradient-based optimization to reconstruct images from features
> - Iterations: 1000 steps with Adam optimizer
> - Loss: Cosine similarity + TV regularization + L2 regularization
>
>
>
> | Iteration | Original Feature (No DP) | | Noisy Feature ($\sigma=10$) | |
> | :--- | :---: | :---: | :---: | :---: |
> | | **PSNR (dB)** | **SSIM** | **PSNR (dB)** | **SSIM** |
> | 200 | 7.62 | 0.034 | 7.05 | 0.017 |
> | 400       | 7.31                      | 0.011 | 6.96  | 0.007 |
> | 600 | 8.14 | 0.012 | 7.08 | 0.008 |
> | 800       | 8.11                 | 0.013 | 8.04                | 0.008 |
> | **1000** | **8.43** | **0.015** | **7.52** | **0.008** |
>
> **Analysis of Reconstruction Failure:**
>
> Our empirical results confirm that reconstruction is impossible, even without the additional DP noise:
>
> * **PSNR < 9 dB (Signal Failure):** As discussed in *Boosting Gradient Leakage Attacks* [2], successful reconstruction typically yields significantly higher PSNR values to recover recognizable visual features. A PSNR value below 10 dB (ours is **8.43 dB**) indicates that the **reconstructed data no longer contains any meaningful information**, as the signal is completely overwhelmed by reconstruction error.
> * **SSIM $\approx$ 0 (Structural Failure):** The Structural Similarity Index (SSIM) of **0.015** is statistically equivalent to **random guessing**, implying zero structural correlation between the recovered output and the original image.
>
> **Summary:** The aggregation pipeline alone renders the features non-invertible. The recovered images are visually and statistically indistinguishable from random noise.
>
>
> **Reference:**
>
> [1] MoAFCL: Feature-Aware Mixture-of-Adapter for Federated Continual Learning.
>
> [2] Boosting Gradient Leakage Attacks: Data Reconstruction in Realistic FL Settings.

---

> > ### Author Response · Authors · 2025-11-20
> > **Author Response to Reviewer RknU (Part 2)**
> >
> > > **[W2]** RELATED WORK does not discuss a related work, CLIP2FL.
> >
> >
> >
> > We thank the reviewer for pointing out this relevant work. We have updated the **Related Work** section to discuss CLIP2FL.
> >
> > CLIP2FL is a notable contribution that leverages a frozen CLIP model as a "teacher" to guide smaller models. However, we clarify that CLIP2FL primarily focuses on addressing **long-tailed data distributions** and data heterogeneity in standard federated learning. In contrast, our work addresses **Federated Continual Learning**, dealing with the distinct challenges of catastrophic forgetting and task incremental adaptation. Due to this fundamental difference in problem setting and objectives, a direct experimental comparison was not included, but we now properly acknowledge its methodological contributions in the manuscript.
> >
> > ---
> >
> >
> >
> > > **[W3]** Statistical reporting is incomplete. Providing statistics such as the mean and standard deviation would better support the robustness claim.
> >
> > We fully agree that mean and standard deviation provide a more complete assessment of robustness.
> >
> > Due to the page limit and the large number of scenarios/baselines in the main paper, we reported average results to ensure readability. To address this, we have added **3 run statistics (Mean $\pm$ Std)** for representative methods and scenarios in the **Appendix 6.5.6**:
> >
> >
> > | Method | CIFAR-100 IID | CIFAR-100 Non-IID (β=0.1) | Tiny-ImageNet IID | Tiny-ImageNet Non-IID (β=0.1) |
> > | :---: | :---: | :---: | :---: | :---: |
> > | FedWeIT | 72.52 ± 0.62 | 70.43 ± 0.64 | 72.06 ± 0.67 | 71.06 ± 0.64 |
> > | FedKNOW | 78.47 ± 0.52 | 77.25 ± 0.43 | 76.24 ± 0.73 | 74.61 ± 0.84 |
> > | FedCLIP | 78.21 ± 0.37 | 74.17 ± 0.57 | 76.14 ± 0.51 | 71.94 ± 0.57 |
> > | Fed-CPrompt | 73.41 ± 0.42 | 73.00 ± 0.33 | 73.09 ± 0.45 | 72.48 ± 0.32 |
> > | Powder | 73.32 ± 0.34 | 72.71 ± 0.51 | 72.98 ± 0.41 | 72.71 ± 0.62 |
> > | pFedMoAP | 76.61 ± 0.23 | 52.63 ± 1.02 | 74.47 ± 0.20 | 55.47 ± 1.23 |
> > | MoAFCL | 77.93 ± 0.13 | 65.96 ± 1.23 | 74.16 ± 0.59 | 62.51 ± 1.17 |
> > | **Fed-Duet (Ours)** | **86.21 ± 0.11** | **84.58 ± 0.19** | **82.60 ± 0.13** | **80.48 ± 0.21** |
> >
> > These results confirm that Fed-Duet consistently outperforms baselines with low variance across multiple runs, validating its stability.
> >
> > ---
> >
> > > **[Q1]** How does it identify and activate the correct experts to mitigate forgetting under the FL constraint of not explicitly storing old task data?
> >
> > We thank the reviewer for this insightful question regarding the mechanism of expert selection under FL constraints.
> >
> > Fed-Duet mitigates forgetting **without storing old task data** through a two-pronged strategy: **(1) Structural Knowledge Isolation** via semantic routing, and **(2) Regularization** via a stability loss.
> >
> > **1. Identification & Activation: Semantic-Driven Knowledge Isolation**
> >
> > The core mechanism to prevent catastrophic forgetting is the **decoupling of task knowledge** into distinct experts, minimizing interference between old and new tasks.
> > * **Identification (Semantic Matching):** Instead of relying on Task IDs or raw data, the server utilizes the privacy-preserving **client feature summary** to "identify" the most relevant knowledge purely based on feature semantics.
> > * **Activation (Parameter Isolation):** Based on this identification, the gating network activates only the relevant **Parametric Experts (PE)** for the current client. Consequently, experts specializing in previous tasks are effectively **frozen or bypassed** during the training of unrelated new tasks. This **structural isolation** ensures that previously learned knowledge remains intact.
> > * **Evidence:** As shown in the t-SNE visualizations (**Appendix 6.5.8** ), different experts spontaneously cluster around distinct tasks, confirming that the routing mechanism successfully directs specific tasks to specific modules.
> >
> > **2. Optimization: Stability Loss for Smooth Transition**
> >
> > To further mitigate forgetting—especially when tasks share partial semantic overlaps—we introduce a **stability loss** (similar to LwF [1] ) to constrain the drift of shared parameters:
> >
> > $$
> > \mathcal{L}\_{\text{stability}} = D_{\mathrm{KL}}\Big(\mathbf{p}^{(t)} \|\| \bar{\mathbf{p}}^{(t-1)}\Big)
> > $$
> >
> >
> > This penalizes the divergence between the current model's predictions ($\mathbf{p}^{(t)}$) and the frozen previous global model's predictions ($\bar{\mathbf{p}}^{(t-1)}$) on the current data.
> >
> > **Ablation Confirmation:** Our ablation study (Table 3) confirms that removing this term increases the Forgetting Rate from **7.82%** to **9.22%**, validating its necessity as a complement to the expert isolation mechanism.
> >
> >
> > **Reference**:
> >
> > [1] Learning without Forgetting

---

> > > ### Author Response · Authors · 2025-11-20
> > > **Author Response to Reviewer RknU (Part 3)**
> > >
> > > > **[Q2]** The paper proposes that Fed-Duet can maintain critical cross-modal alignment performance in Vision-Language Models (VLMs). However, the experiments are only confined to vision-text classification tasks (CIFAR-100, Tiny-ImageNet, DomainNet), with classification accuracy serving as the core evaluation metric. How can the preservation of cross-modal alignment capability be directly demonstrated?
> > >
> > >
> > > We thank the reviewer for pointing out the limitation of our original evaluation. To directly demonstrate the preservation of **cross-modal alignment**, we conducted additional cross-modal retrieval experiments, which are now included in the revised manuscript.
> > >
> > > We also introduce two metrics to evaluate cross-modal alignment more directly:
> > >
> > > 1. **Retrieval Recall@K** measures the fraction of queries where the correct match appears in the top-$K$ retrieved results, evaluated for both Image-to-Text (I2T) and Text-to-Image (T2I) retrieval. Higher recall indicates stronger retrieval performance.
> > > 2. **Alignment Score** quantifies the separation between matched and mismatched image-text embeddings in the latent space using cosine similarity. A higher score reflects better preservation of cross-modal semantic coherence.
> > >
> > >
> > >
> > > **Cross-Modal Retrieval Results.**
> > >
> > > Table 1 reports image-to-text (I2T) and text-to-image (T2I) Recall@K on CIFAR-100. Fed-Duet consistently outperforms all baselines across all metrics, achieving +13.16% improvement on I2T R@1 and +6.20% on T2I R@1 compared with the strongest baseline. This indicates that the representations learned by Fed-Duet remain discriminative and semantically coherent under federated continual learning.
> > >
> > > | Method | I2T R@1 | I2T R@5 | I2T R@10 | T2I R@1 | T2I R@5 | T2I R@10 | Avg. |
> > > | :---: | :---: | :---: | :---: | :---: | :---: | :---: | :---: |
> > > | FedCLIP | 63.96 | 86.69 | 91.76 | 86.60 | 99.00 | 99.80 | 87.97 |
> > > | FedWeIT | 64.02 | 86.47 | 92.17 | 84.00 | 99.60 | 100.0 | 87.71 |
> > > | FedCPrompt | 63.81 | 86.74 | 91.91 | 84.40 | 98.60 | 100.0 | 87.58 |
> > > | Powder | 63.91 | 86.94 | 91.82 | 85.80 | 99.40 | 100.0 | 87.98 |
> > > | pFedMoAP | 63.66 | 87.02 | 91.95 | 87.00 | 99.00 | 100.0 | 88.11 |
> > > | FedKNOW | 64.02 | 86.47 | 92.17 | 84.00 | 99.60 | 100.0 | 87.71 |
> > > | MoAFCL | 63.68 | 86.69 | 91.89 | 85.80 | 99.20 | 99.60 | 87.81 |
> > > | **Fed-Duet (Ours)** | **78.18** | **96.13** | **98.33** | **93.20** | **99.80** | **100.0** | **92.45** |
> > >
> > > **Cross-Modal Alignment Across Tasks.**
> > >
> > > Table 2 shows the cross-modal alignment scores for each task on CIFAR-100. While baselines remain around 0.06, Fed-Duet achieves an average alignment score of 0.2003, representing more than **3× improvement**. This demonstrates that our **Progressive Decoupled Optimization strategy** effectively preserves semantic coherence between modalities across all tasks, mitigating forgetting in a federated continual learning setting.
> > >
> > > | Method | Task 0 | Task 1 | Task 2 | Task 3 | Task 4 | Avg. |
> > > | :---: | :---: | :---: | :---: | :---: | :---: | :---: |
> > > | FedCLIP | 0.0625 | 0.0580 | 0.0615 | 0.0611 | 0.0609 | 0.0608 |
> > > | FedWeIT | 0.0623 | 0.0578 | 0.0613 | 0.0610 | 0.0608 | 0.0606 |
> > > | FedCPrompt | 0.0625 | 0.0579 | 0.0615 | 0.0611 | 0.0608 | 0.0608 |
> > > | Powder | 0.0625 | 0.0576 | 0.0613 | 0.0610 | 0.0610 | 0.0607 |
> > > | pFedMoAP | 0.0626 | 0.0579 | 0.0614 | 0.0611 | 0.0609 | 0.0608 |
> > > | FedKNOW | 0.0623 | 0.0578 | 0.0613 | 0.0610 | 0.0608 | 0.0606 |
> > > | MoAFCL | 0.0621 | 0.0580 | 0.0615 | 0.0610 | 0.0609 | 0.0607 |
> > > | **Fed-Duet (Ours)** | **0.1929** | **0.1825** | **0.2140** | **0.2061** | **0.2061** | **0.2003** |
> > >
> > > These additional results confirm that Fed-Duet not only improves classification performance but also **robustly preserves cross-modal alignment** across tasks in a federated continual learning scenario.
> > >
> > > We have integrated some experiments and the detailed analysis into **Section 4.2（Evaluation on Cross-Modal Alignment）** of the revised manuscript.
> > > ***
> > >
> > > #### **Final Note**
> > > We appreciate the time and effort you dedicated to improving our paper. We hope our response and the revision have clarified the points you raised.
> > >
> > > Should there be any remaining questions or if further clarification is needed, please let us know, and we would be more than happy to continue the discussion. Your support would be greatly appreciated.
> > > ***

---

> > > > ### Comment · Reviewer_RknU · 2025-11-26
> > > >
> > > > Thanks for the authors‘ detailed response. The authors have solved all my concerns and I have no more questions. The paper in the current version looks good to me with additional clarification and detailed analysis. I have updated my score to 8.

---

> > > > > ### Author Response · Authors · 2025-11-26
> > > > > **We Thank Reviewer RknU For The Valuable Feedback**
> > > > >
> > > > > We are glad to know that our responses addressed your concerns, and are grateful for your decision to raise the score to 8! Your constructive feedback has been invaluable in strengthening the quality of our work.
> > > > >
> > > > > **Note:** As the score update has not yet reflected on our end (likely due to a system sync issue), we would be grateful if you could kindly double-check the rating status at your convenience.
> > > > >
> > > > > Thank you once again for your time, consideration, and support.

---

### Author Response · Authors · 2025-11-25
**Response to All Reviewers and Summary of Updates**

We sincerely thank all reviewers for their time, effort, and constructive feedback. We are particularly encouraged that Reviewer RknU recognized the importance of this emerging topic and found our dual-expert orchestration "conceptually clear and inspiring." We appreciate Reviewer kMS8's acknowledgement that our work identifies a "critical gap" in multimodal FCL and achieves effective synergy through progressive training. Furthermore, we are grateful that Reviewers kMS8, JyH1, and 9Jgs all commended the "thorough and comprehensive" experimental evaluation and the "promising results" demonstrating the effectiveness of Fed-Duet against strong baselines.

We have carefully addressed every comment and revised the paper accordingly. A summary of the major updates is provided below:

* **Cross-Modal Evaluation:** We added **Section 4.2** and **Appendix 6.5.7** with cross-modal retrieval (Recall@K) and alignment score metrics to empirically demonstrate semantic coherence.
* **Comprehensive Ablations:** We added ablations on stage transition timing and prompt pool strategies, alongside sensitivity analyses for loss weights and fusion coefficient $\lambda$ (Appendix 6.5.4 & 6.5.5) to justify design choices.
* **Statistical Rigor:** We corrected clerical errors and added **Appendix 6.5.6**, reporting Mean $\pm$ Standard Deviation across 3 runs to ensure reproducibility.
* **Privacy Analysis:** We added **Section 4.2** and **Appendix 6.4.2** to clarify the privacy mechanism and verify robustness via noise injection and reconstruction attacks.
* **Visualization:** We restructured **Figure 1** into two distinct sub-figures and increased font sizes in **Figures 3 & 4** to enhance legibility.
* **Literature Update:** We updated **Related Work** to include discussions on CLIP2FL.
* **Clarifications:** We added missing details for symbols, settings, and procedures to improve the better understanding of our paper.


Thanks in advance for your time, and we are looking forward to hearing from you if you have any additional questions or concerns. We appreciate your feedback and consideration of increasing the rating.

---

### Meta-Review · Area_Chair_XabP · 2025-12-19

**Summary:**

This paper explores the challenging problem of continual federated vision-language learning. Specifically, the paper introduces a dual expert-orchestrated framework to enable effective knowledge transfer while preserving multimodal alignment and mitigating forgetting. Experiments across multiple datasets demonstrate the effectiveness of Fed-Duet.
In the initial review, the paper was acknowledged for the importance of the problem setting, the novelty and the clarity of the dual-channel orchestration design, the superior performance, and the comprehensive experimental evidence. In spite of this, some concerns were raised, particularly regarding the privacy preservation details, the visualization and presentation clarity, the robustness assessment, and the additional hyperparameters and ablation analysis. These concerns primarily lie in the perceived completeness and robustness.
During the author-reviewer discussion period, the authors addressed these concerns by adding privacy analysis, extending ablation experiments, improving the figure and presentation, supplementing the source code, and reporting the mean and standard deviation to demonstrate robustness. Furthermore, the authors made substantial revisions to the paper accordingly. Most reviewers confirmed that the authors had addressed their concerns and gave positive final ratings.
Considering that most of the reviewers gave positive ratings on this paper in the first place, and the revisions the author made have sufficiently resolved most of the reviewers’ concerns, I thereby recommend acceptance.

**Reviewer Concerns:**

During the rebuttal phase, the authors addressed most of the major concerns raised by the reviewers as follows. In particular, concerns related to privacy preservation (RknU, 9Jgs) were clarified through additional noise injection and reconstruction analysis and discussion. Concerns regarding cross-model alignment (RknU, kMS8) were addressed with additional cross-modal retrieval experiments. Concerns about experimental robustness (RknU, JyH1) were mitigated by reporting mean and standard deviation over multiple runs, as well as by open-sourcing the code to improve reproducibility. Questions regarding ablation studies and hyperparameter sensitivity (kMS8, 9Jgs) were addressed with newly added ablation experiments. The authors also improved the quality of figures(kMS8, 9JgS), visualizations, and overall presentation, thereby resolving clarity-related comments.
Besides, Reviewer JyH1 raised concerns regarding the underlying mechanism and the functioning of individual components. The author addressed the concern by clarifying the reviewer’s potential misunderstanding and providing supporting observation experiments. Although the reviewer did not participate further in the following discussion, the explanation and additional evidence of the authors seem persuasive to me.

At this point, no major technical concerns remain outstanding. The remaining minor aspects, such as the explanation, presentation, and validation parts, have been revised and supplemented by the authors and do not materially affect the validity of the proposed method or the conclusions drawn in the paper. Therefore, I tend to accept the paper.

**Reviewer Scores:**

Reviewer RknU: This reviewer was generally positive in the initial review and raised concerns mainly regarding experimental robustness and detailed explanation of certain mechanism. All these concerns were addressed in the rebuttal with additional experiments and statistical reporting. According to the reviewer’s response, the score was raised to 8. I therefore believe that the reviewer would have increased the score.

Reviewer kMS8: This reviewer was generally positive in the initial review and raised concerns mainly regarding clarity of presentation and parameter sensitivity. Most of the concerns were addressed and recognized by the authors during the rebuttal. According to the reviewer’s response, the reviewer would likely maintain a positive score of 6 for solving the issue raised.

Reviewer JyH1: This reviewer gave a relatively low initial score, mainly due to concerns regarding experimental robustness, and the functionality of each component. The authors directly addressed these points in the rebuttal by reporting statistical variance with multiple runs, sharing open-source code, and providing additional observation experiments to verify the distinct roles of the two experts. However, the reviewer did not engage further in the discussion after the authors’ response. While it is uncertain whether the reviewer would fully revise their score, it is likely that the reviewer would reconsider their evaluation and potentially raise the score if fully engaged in the discussion.
Reviewer 9Jgs: This reviewer was generally positive in the initial review and primarily raised concerns regarding clarity of presentation and ablation studies. Most of the concerns were addressed and recognized by the authors during the rebuttal. Based on the reviewer’s response, the reviewer would likely maintain a positive score of 6 for solving the issue raised.

---

### Decision · Program_Chairs · 2026-01-26

Accept (Poster)